# High-performance piezoelectric energy harvesting in amorphous perovskite thin films deposited directly on a plastic substrate

Ju Han [1], Sung Hyun Park[1], Ye Seul Jung[1] & Yong Soo Cho [1] ✉

Most reported thin-film piezoelectric energy harvesters have been based on cantilever-type crystalline ferroelectric oxide thin films deposited on rigid substrates, which utilize vibrational input sources. Herein, we introduce flexible amorphous thin-film energy harvesters based on perovskite $CaCu_3Ti_4O_{12}$ (CCTO) thin films on a plastic substrate for highly competitive electromechanical energy harvesting. The room-temperature sputtering of CCTO thin films enable the use of plastic substrates to secure reliable flexibility, which has not been available thus far. Surprisingly, the resultant amorphous nature of the films results in an output voltage and power density of ~38.7 V and ~2.8 × 10$^6$ µW cm$^{-3}$, respectively, which break the previously reported record for typical polycrystalline ferroelectric oxide thin-film cantilevers. The origin of this excellent electromechanical energy conversion is systematically explored as being related to the localized permanent dipoles of $TiO_6$ octahedra and lowered dielectric constant in the amorphous state, depending on the stoichiometry and defect states. This is the leading example of a high-performance flexible piezoelectric energy harvester based on perovskite oxides not requiring a complex process for transferring films onto a plastic substrate.

Piezoelectric energy harvesters are commonly constructed in the form of thin-film cantilevers based on well-known perovskite oxides, such as $Pb(Zr,Ti)O_3$ (PZT)[1–5], $(K,Na)NbO_3$[6,7], and $(Bi,Na)TiO_3$[8,9]. Because the piezoelectricity of these oxides effectively appears in the crystalline state, processing at a high temperature, typically greater than 700 °C, is critical for the crystallization of thin films, usually using a rigid Si substrate[10–12]. For flexible systems, however, such high-temperature processing is not suitable because of its incompatibility with most polymer-based substrates. Highly flexible harvesters offer the merits of using a variety of mechanical input sources, such as bending, stretching, and warping, for diverse electromechanical conversion devices, in contrast with the vibration-based harvesting of thin-film cantilevers[13–17]. Considerable efforts have been devoted to incorporating high-quality polycrystalline thin films into flexible systems by adopting complicated and size-limited processes to transfer the separately prepared thin films. These transfer methods include the

exfoliation of annealed films from a mica substrate[18], wet-etching of Si substrates[19], laser lift-off of perovskite films[20], and stress-driven exfoliation of polycrystalline films from Ni[21].

Herein, we introduce a flexible piezoelectric energy harvester based on perovskite $CaCu_3Ti_4O_{12}$ (CCTO) thin films deposited directly onto a plastic substrate without a post-annealing process, which demonstrates excellent power generation under the bending operation. Polycrystalline CCTO has not been recognized as a typical piezoelectric material but is known to possess an ultrahigh dielectric constant $\varepsilon_r$ of >10$^4$ at low frequencies, and the temperature dependence of its $\varepsilon_r$ value remains stable between 100 and 600 K[22–24]. The origin of the colossal dielectric permittivity is understood as being related mainly to the Maxwell–Wagner polarization at low frequencies, in which the build-ups of ionized charges occur at the electrode–sample interfaces and across the grain boundaries[25–27]. The structural complexity associated with the defect chemistry and

[1]Department of Materials Science and Engineering, Yonsei University, Seoul 03722, Korea. ✉e-mail: ycho@yonsei.ac.kr

the mixed chemical states of cations depends on the sample preparation conditions[28,29]. Particularly, an internal barrier layer capacitor model based on n-type semiconducting grains and insulating grain boundaries was suggested as a unique characteristic of the intrinsic CCTO material[26,27]. Charge transport is inhibited by a potential barrier at the grain boundary, inducing a larger polarization with accumulated charges. The interfacial (or space-charge) polarization caused by the large defect-dipoles across the interfaces or grain boundaries primarily contributes to the large $\varepsilon_r$, which is very unusual because most other perovskite-based devices rely on the dipolar polarization mechanism responsible for strong ferroelectricity. It may be valuable to mention that CCTO itself also possesses ferroelectricity, potentially attributable to the off-centered displacements of Ti ions and the tilting of the $TiO_6$ octahedra in the long-range ordered ferroelectric state, as reported elsewhere[30,31].

However, the polarization mechanism of the amorphous phase of CCTO should be different. We previously reported the amorphous state of CCTO thin films sputtered at room temperature, with a promising $\varepsilon_r$ reaching ~192, which remains one of the highest values for any amorphous oxides[32]. The high $\varepsilon_r$ value in the amorphous state was successfully exploited in thin-film transistors consisting of all-amorphous layers, with notable transistor performance[33]. To explain the unusual permittivity of amorphous CCTO films, defect-dependent chemical states, and short-range grain boundaries were suggested to be the main contributors, as supported by experimental results[32]. Interestingly, there have been a few reports on dipolar polarization in cases of amorphous perovskite oxide films, e.g., amorphous $BaTiO_3$[34] and $SrTiO_3$[35,36], which experimentally demonstrated that the magnitude of Ti off-centered displacement was higher than the case of crystalline counterparts owing to the weak orientational ordering of $TiO_6$ octahedra in the amorphous state. Such an amorphous state concerning the dipolar polarization was explained based on the random network of local bonding units (RN-LBU) theory, proposing that piezoelectricity can exist in structures that lack the spatial periodicity inherent for ionic crystals but are composed of polar units with directional ordering in short-range scale. However, the polarity may not be greatly effective in the disordered structure, partially because the neighboring octahedra can be connected via edges and faces (not via apex-to-apex). In addition, there are rare reports on effective piezoelectric coefficients measured by piezoresponse force microscopy (PFM) in the cases of amorphous perovskite oxide films, such as 16 pm $V^{-1}$ for 300-nm-thick $KNbO_3$ film[37] and 10 (±25%) pm $V^{-1}$ for 100-nm-thick $SrTiO_3$ film[36], even though the PFM results may not represent intrinsic values owing to the limitation of the PFM instrument itself[38,39].

This study introduces flexible amorphous CCTO thin films for high-performance power-generation devices utilizing bending-driven electromechanical energy conversion. To the best of our knowledge, amorphous perovskite thin films have not been explored thus far for piezoelectric power-generation applications. The energy-harvesting performance was excellent despite the amorphous nature of the thin films; peak output values of ~38.7 V, ~413 μW, and ~2.8 × 10⁶ μW cm⁻³ which are the record-high values among reported energy-harvesting devices based on typical polycrystalline-oxide-thin-film cantilevers utilizing mechanical vibrations. The origin of such extraordinary performance is explored in conjunction with potential electromechanical mechanisms based on supporting evidence for the altered chemical states and polarization. We believe that the current example of preparing devices composed of amorphous piezoelectric thin films without a complicated transfer process will be greatly useful for other flexible devices that demand high permittivity and piezoelectricity.

## Results
### Characteristics of amorphous CCTO thin films
CCTO thin films were deposited at room temperature on polyethylene naphthalate (PEN) substrates by RF magnetron sputtering from a stoichiometric $CaCu_3Ti_4O_{12}$ target prepared by a solid-state reaction. Two main parameters, the film thickness and oxygen partial pressure $pO_2$, were modulated to optimize the harvesting performance. The cross-sectional transmission electron microscopy (TEM) images in Fig. 1a show distinct layers corresponding to the CCTO thin films deposited directly onto the Pt (~100 nm thick)-coated PEN substrate, where films with different thicknesses of up to ~497 nm exhibit a clear interface between the dielectric layer and Pt. The films were identified as amorphous regardless of the film thickness, as evidenced by the condensed diffraction patterns shown as insets in Fig. 1a. Figure 1b presents the X-ray diffraction (XRD) patterns of the as-deposited films deposited at different $pO_2$ from 1.8 to 4.0 mTorr. For reference, the XRD pattern of the Pt/PEN substrate is included. All films were identified as amorphous in the patterns, which only exhibit peaks corresponding to the Pt electrode, irrespective of the oxygen partial pressure. Note that the crystallization of the sputtered CCTO thin films typically requires an annealing temperature of approximately 600 °C[32,40].

Changes in the chemical states of the films were examined by high-resolution X-ray photoelectron spectroscopy (XPS), as shown in Fig. 1c, which shows the spectra of the Cu $2p_{3/2}$, Ti $2p_{3/2}$, and O $1s$ states for the CCTO thin films deposited at different $pO_2$. The spectra were calibrated using the C $1s$ peak (284.8 eV) as the internal standard. All curves were fitted using the Lorentzian–Gaussian functions. The XPS spectra of the Cu $2p_{3/2}$ state in the binding energy region of 927–939 eV suggest the coexistence of $Cu^{1+}$ and $Cu^{2+}$ valence states. Multiple peaks for $Cu^{2+}$ and a single $Cu^{1+}$ peak are clearly visible. For the Ti $2p_{3/2}$ states, two fitted peaks at ~456.2 and ~457.3 eV for $Ti^{3+}$ and $Ti^{4+}$ ions, respectively, were distinctly observed, with relative peak areas depending on $pO_2$. The mixed valence states of the cations are well recognized in CCTO materials, which induce structural complexity with the defect chemistry[29,31]. The relative ratio of each cation was calculated from the peak areas to trace the changes with $pO_2$, as shown in the plots in Fig. 1d. As expected, the relative ratios of both $Cu^{2+}/Cu^{1+}$ and $Ti^{4+}/Ti^{3+}$ increased with $pO_2$, which indicates the increasing oxidation of the $Cu^{1+}$ and $Ti^{3+}$ cations with the supply of more oxygen. For example, the $Ti^{4+}/Ti^{3+}$ ratio rose from 1.04 at 1.8 mTorr to 1.64 at 4.0 mTorr. $Ti^{4+}$ is known to reduce to $Ti^{3+}$ as the electrons released from oxygen vacancies are accepted by the conduction band of $Ti^{4+}$, which is responsible for the n-type conductivity of the grains in CCTO[41]. In the case of oxygen (Fig. 1c), three major peaks were observed at binding energies of 528–533 eV: the peak at 529.7 eV corresponding to the $O^{2-}$ ions in the perovskite lattice ($O_L$), another at 531.3 eV from the oxygen vacancies ($O_V$), and one more at 531.9 eV caused by adsorbed oxygen ($O_a$) (or OH species) on the film surface[42,43]. With the increasing $pO_2$, the normalized area of $O_V$ was found to decrease relative to the area of $O_L$, as shown in the plot of $O_V/O_L$ versus $pO_2$ (Fig. 1d), indicating that the supply of more oxygen produced fewer oxygen vacancies. The decreased oxygen vacancies contributed to the higher valence states of cations to compensate for the charge imbalance in the amorphous structure[32,40]. The stoichiometry of the amorphous films was estimated based on the XPS analysis, as presented in Supplementary Table 1, where the changes in atomic ratios with $pO_2$ are listed. The higher $pO_2$ changed the atomic ratios toward the stoichiometric ratios of crystalline CCTO. As a result, nearly perfect stoichiometric ratios of 1.00:3.01:3.99:12.00 for Ca:Cu:Ti:O were achieved for the film processed at the highest $pO_2$ of 4.0 mTorr, indicating that more supply of oxygen is critical in reaching the stoichiometric ratios.

### Piezoelectric energy harvesting of amorphous CCTO thin films
The amorphous state of the CCTO film was explored to define the electromechanical energy-harvesting characteristics with the variations in the film thickness and $pO_2$. As shown schematically in Fig. 2a, the harvester structure was PEN/indium tin oxide

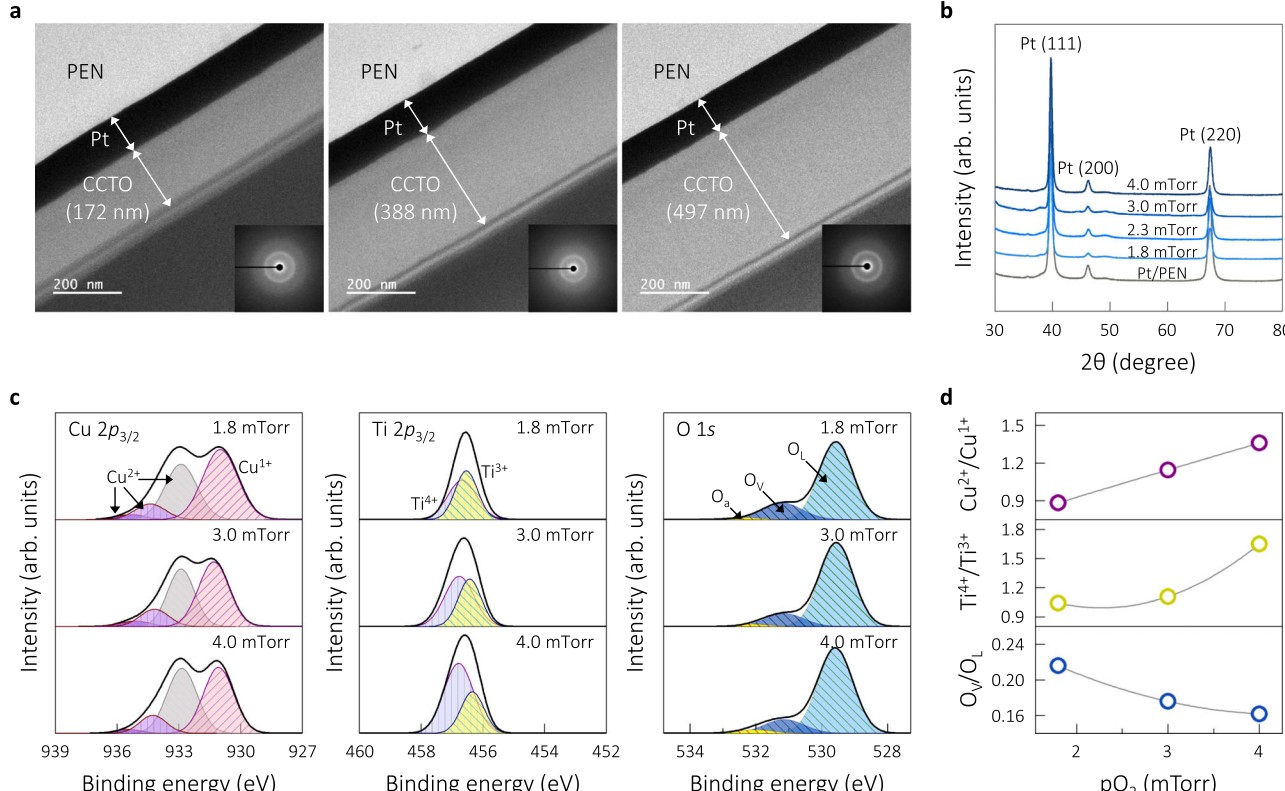

**Fig. 1 | Characteristics of amorphous CaCu$_3$Ti$_4$O$_{12}$ thin films. a** TEM images of the CCTO thin films on Pt/PEN substrates, which were sputtered at 4.0 mTorr while changing the deposition time to vary the thickness (inset: condensed diffraction patterns). **b** XRD patterns of the thin films deposited at different pO$_2$ (with the Pt/PEN pattern for reference), suggesting no crystalline perovskite phase formed during sputtering. **c** XPS spectra of the CCTO thin films deposited at pO$_2$ of 1.8, 3.0, and 4.0 mTorr for the binding energy regions corresponding to the Cu 2$p_{3/2}$, Ti 2$p_{3/2}$, and O 1$s$ states, with the fitted curves of the multiplet Cu$^{2+}$, Cu$^{1+}$, Ti$^{4+}$, Ti$^{3+}$, O$_a$ (adsorbed oxygen), O$_V$ (oxygen vacancy), and O$_L$ (oxygen in lattice) peaks in the corresponding states. **d** Changes in the relative ratios of Cu$^{2+}$/ Cu$^{1+}$, Ti$^{4+}$/Ti$^{3+}$, and O$_V$/O$_L$ with the increasing pO$_2$. Source data are provided as a Source data file.

(ITO)/polydimethylsiloxane(PDMS)/CCTO/Pt/PEN, wherein the top and bottom PEN layers had different thicknesses of ~50 and ~125 μm, respectively. The thin layer of PEN on top was intended to passivate the active films, with the effect of positioning the neutral plane in the lower part of the device structure to provide sufficient tensile strain in the CCTO films during the bending-driven harvesting measurement. The procedure for estimating the position of the neutral plane is described in Supplementary Note 1 with the schematic illustration of Supplementary Fig. 1, along with the calculated results of the bending strain $s_b$ depending on the film thickness (Supplementary Table 2). The $s_b$ values ranged from 0.45% to 0.77%, as determined by different levels of the bending curvature. For example, an $s_b$ value of 0.45% was obtained with a curvature of 7.03 mm, while a larger $s_b$ of 0.77% was attained for a curvature of 4.06 mm. The harvesting performance was recorded for various oxygen partial pressures, film thicknesses, bending strains, and bending frequencies. Figure 2b, c shows the effects of different pO$_2$ on the piezoelectric energy-harvesting performance for the ~497-nm-thick CCTO film, which was measured at a bending strain of 0.77% and a bending frequency of 3.10 Hz. Evidently, the harvesting performance was largely dependent on pO$_2$, as will be discussed later in terms of the origin of these enhancements. The outputs reached ~22.0 V and ~906 nA for the highest pO$_2$ of 4.0 mTorr, which are quite impressive relative to the low values of ~4.5 V and ~273 nA achieved with the lowest pressure of 1.8 mTorr. It should be mentioned that the intermediate PDMS layer was used to generate more stable energy harvesting performance as seen in Supplementary Fig. 2 where more consistent peak-to-peak output values were created with the incorporation of PDMS layer compared to the results produced by the CCTO layer without PDMS. The harvesting performance of the PDMS layer alone was trivial,

as demonstrated in Supplementary Fig. 3, as it generated an output voltage and current of only ~0.05 V and ~17 nA, respectively.

The thickness of the CCTO films also significantly affected the energy-harvesting performance, as demonstrated by the variations in the peak values in Fig. 2d, with the original plots shown in Supplementary Fig. 4. A thicker film resulted in stronger energy-harvesting performance. Compared to the thinnest film (~86 nm), the thickest film (~497 nm) demonstrated ~368% and ~597% higher peak voltage and current, respectively. Harvesting values typically increase with the thickness of piezoelectric thin films because the effect of the substrate interface becomes less influential as the material characteristics approach those of the bulk[17,44–46]. The effects of increasing the bending strain from 0.45% to 0.77% and bending frequency from 0.86 to 3.50 Hz on the energy-harvesting performance are shown in Supplementary Fig. 5, with the plotted example of the strain case in Fig. 2e. The optimal bending conditions were 0.77% and 3.10 Hz. The polarity-switching behavior of the harvester, as illustrated in Supplementary Fig. 6 with forward or backward connection, suggests that the harvesting outcomes were produced by the CCTO films. We also confirmed the stability of the harvesting performance by continuing bending up to 11,000 cycles, as shown in Fig. 2f, where consistently stable output values were attained over the extended cycles at 2.73 Hz for the ~497-nm-thick film. Supplementary Fig. 7 shows another case of harvesting stability at a higher bending frequency of 2.90 Hz for the identical sample, confirming that the physical integrity is well preserved during repetitive bending operations. The chemical stability of the amorphous CCTO films was evaluated in terms of changes in the chemical states of ions after exposing the films to an ambient atmosphere for 15 days as compared in the XPS spectra of Supplementary

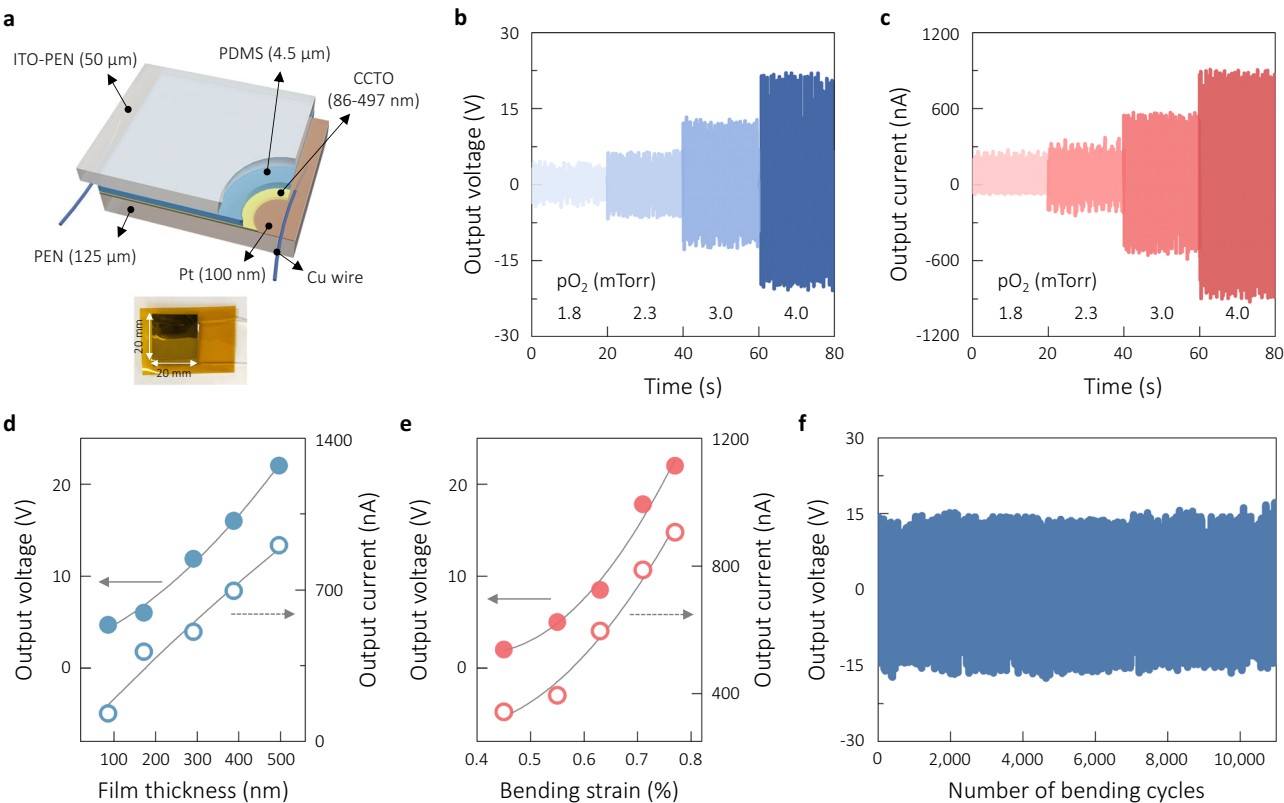

**Fig. 2 | Performance of piezoelectric energy harvesting. a** Schematic of the flexible piezoelectric energy harvester based on PEN/ITO/PDMS/CCTO/Pt/PEN and a photograph of an actual harvester sample sealed with a PI tape on both sides. **b, c** Output voltage and current measured under the optimal bending operation conditions of a bending strain and frequency of 0.77% and 3.10 Hz, respectively, for the ~497-nm-thick CCTO thin films sputtered at different oxygen partial pressures. **d** Dependence of the harvesting outputs for the CCTO thin films on the CCTO film thickness measured under optimal conditions. **e** Dependence of the harvesting outputs on the bending strain from 0.45% to 0.77%, measured at 3.10 Hz. **f** Stability evaluation of the output voltage over 11,000 cycles for the CCTO thin film deposited at 4.0 mTorr, measured at the bending strain of 0.77% and the bending frequency of 2.73 Hz. Source data are provided as a Source data file.

Fig. 8 before and after the exposure. As expected, no noticeable chemical changes were observed with no degradation in the harvesting performance after the exposure period, as presented in Supplementary Fig. 9. As another effort, the harvesting performance was monitored with the changes in humidity level from 30 to 80 % as seen in Supplementary Fig. 10, suggesting that the optimal harvester device is quite stable at the humid atmosphere.

**Origin of the high piezoelectricity in amorphous CCTO thin films**
The piezoelectricity of the amorphous CCTO films was explored using the analytical methods. Figure 3a represents the images of the piezoresponse (PR) amplitudes obtained by applying an AC bias of 4 V at 40 kHz over a scan area of ~5 μm × ~5 μm on the CCTO films processed at different pO$_2$. The brighter contrast in the PR amplitude images with higher pO$_2$ indicates a larger PR over the entire area, confirming the stronger electromechanical behavior of the films processed at higher pO$_2$. As shown in Supplementary Fig. 11, the distributions of the PR amplitudes broadened toward the higher amplitudes for the films processed with higher pO$_2$. The corresponding PR phase images are available in Supplementary Fig. 12. Figure 3b shows the nonlinear behavior of the PR amplitudes measured with the changed DC bias in the range of −10 V to +10 V after applying an AC bias of 4 V at 40 kHz for the corresponding thin films. Nonlinear behavior with larger amplitudes is evident over the DC bias range, particularly at higher pressures. For the intuitive comparisons in the changed PR amplitudes over the large area, the mapping images of effective piezoelectric coefficient $d_{33,eff}$, which were estimated from the peak amplitude at +10 V, were constructed using 400 data points for the designated area

of ~10 × 10 μm$^2$ as seen in Fig. 3c. The mapping images demonstrate quite consistency over the extended areas, ensuring the positive effect of high pO$_2$ on the piezoresponse. The average $d_{33,eff}$ value increased gradually with the applied oxygen pressure, specifically from 1.73 ± 0.3 pm V$^{-1}$ for the 1.8 mTorr sample to 27.6 ± 4.4 pm V$^{-1}$ for the 4.0 mTorr one, as plotted in Fig. 3d. Note that the average $d_{33,eff}$ may still deviate from intrinsic values due to the limitations of the PFM measurement technique although the comparison of the mapping images clearly supports the pO$_2$-dependent piezoresponse. Note that nearly stoichiometric atomic ratios of CCTO were achieved at the highest oxygen partial pressure of 4.0 mTorr, which is close to the crystalline structure in short range. Piezoelectricity in this amorphous state is believed to come mainly from the dipolar polarization of TiO$_6$ octahedra distributed randomly in a network of dipole units. The evidence of dipolar polarization (or ferroelectricity) has been reported in polycrystalline CCTO thin films processed by various deposition techniques[30−32,47−49].

The potential domain reversal of the 4.0 mTorr sample was examined to ascertain ferroelectricity, as shown in Fig. 3e, where large contrast changes in the PR phase images of the designated boxes were observed by consecutively switching between biases of +10 and −10 V. The apparent ferroelectricity at the high pO$_2$ is believed to be associated with the dipole moments driven by local TiO$_6$ octahedra even in the amorphous state, which become effective with the reduced oxygen vacancies and the changed chemical states of cations with the higher pO$_2$[50,51]. The existence of oxygen vacancies negatively affects polarization by acting as clamping centers against the movement of domain walls[32,52,53]. The increased concentration of Ti$^{4+}$ relative to Ti$^{3+}$ with the

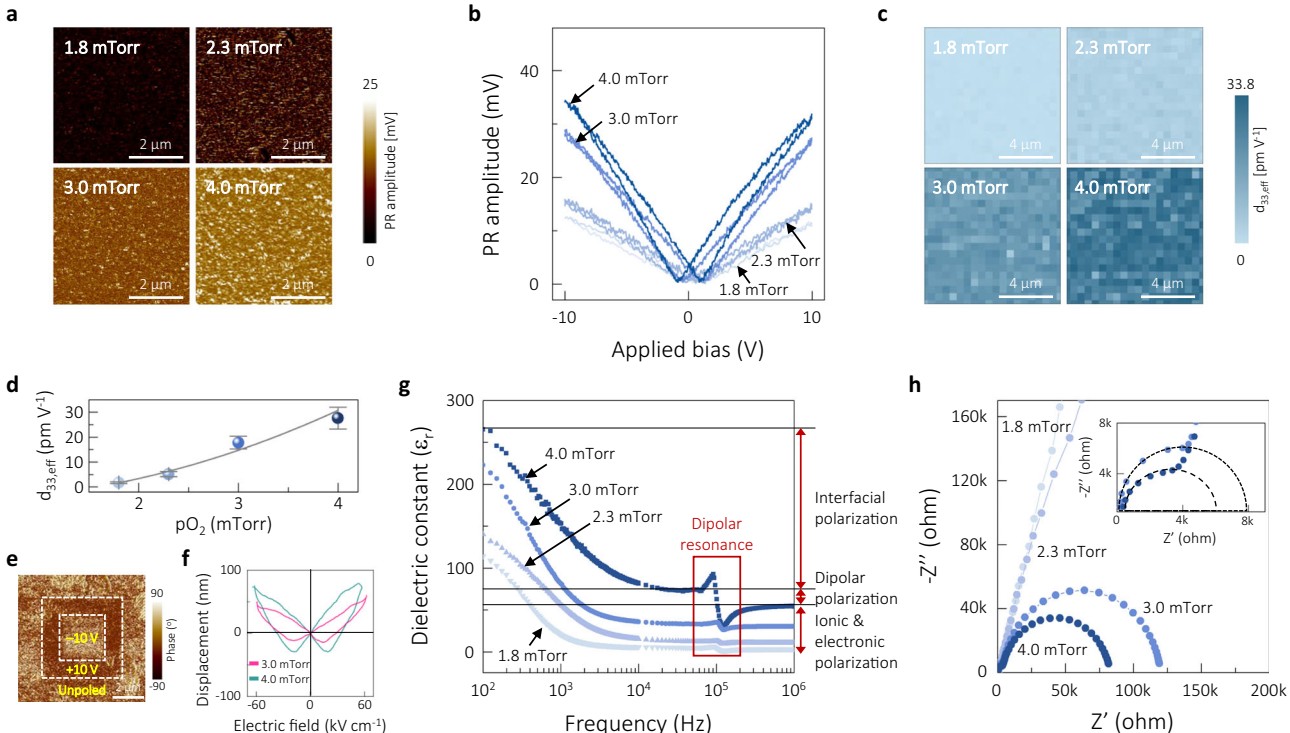

**Fig. 3 | Origin of piezoelectricity in amorphous CCTO films. a** PR amplitude images of the CCTO films deposited at different $pO_2$ recorded with an AC bias of 4 V at 40 kHz. **b** Variations in the PR amplitudes with applied DC bias for the CCTO thin films deposited at different $pO_2$. **c** Mapping images of the calculated $d_{33,eff}$ from the peak amplitude at +10 V for 400 data points over the area of -10 × 10 μm² of each film. **d**. Variations in average $d_{33,eff}$ with $pO_2$ (corresponding to $d_{33,eff}$ values of 1.73 ± 0.3, 5.16 ± 1.1, 17.8 ± 2.6, and 27.6 ± 4.4 pm V⁻¹ in the increasing order). **e** PFM phase image characterized by applying consecutive biases of +10 and −10 V in the designated boxes. **f** Electric displacement versus electric field for the 3.0 and 4.0 mTorr samples, measured by laser interferometry. **g** $pO_2$-dependent changes in dielectric dispersion of amorphous CCTO thin films in the frequency range of -10² to 10⁶ Hz, suggesting the frequency-dependent contributions by different polarization mechanisms (divided into the sections with the parallel lines) with the red box highlighting the dielectric resonance due to the dipolar polarization at -10⁵ Hz. **h** Complex impedance curves of the CCTO thin films deposited at different $pO_2$ (inset: magnified low-impedance region). Source data are provided as a Source data file.

higher $pO_2$ may produce greater polarization because the smaller ionic radius of $Ti^{4+}$ (0.605 Å) than that of $Ti^{3+}$(0.67 Å) may induce a larger dipole moment[30].

To further confirm the existence of piezoelectricity in amorphous CCTO, we made extra efforts to characterize piezoelectric coefficients using two other measurement techniques. Figure 3f shows the electric displacements with an applied electric field for the 3.0 and 4.0 mTorr films on Si, which were recorded using single-beam laser interferometry. The larger non-linear response was found in the case of 4.0 mTorr, resulting in an estimated effective $d_{33}$ of -28.5 pm V⁻¹, which is higher than -18.1 pm V⁻¹ for the 3.0 mTorr sample. Even though there is uncertainty in the values due to the uncompensated bending effect of the substrate (as the limitation of the measurement technique)[39,54], at least it ensures the presence of piezoelectricity in the amorphous state. As a reference, 1.9 μm-thick crystalline PZT films reported effective $d_{33}$ of 90 to 185 pm V⁻¹ as the result of measurement using a double-beam laser interferometer[38]. Further, the effective transverse piezoelectric coefficient $−e_{31,eff}$ was obtained using a commercial 4-point bender unit for the 4.0 mTorr film on Si to verify the anticipated piezoelectric response without the bending effect of substrate. A $−e_{31,eff}$ value of -1.1 C m⁻² was attained for the 4.0 mTorr sample. As references, $−e_{31,eff}$ values of -3.2 C m⁻² for crystalline $(Bi_{0.5}Na_{0.5})TiO_3$-$BaTiO_3$ thin films[8] and -13−16 C m⁻² for crystalline PZT films[2,39] were reported.

We attempted to identify polarization mechanisms of the amorphous CCTO by measuring the dielectric constant $\varepsilon_r$ of the films deposited at different $pO_2$ in the frequency range of 10²–10⁶ Hz, as shown in Fig. 3g. The dielectric dispersion largely depended on the $pO_2$; that is, a higher oxygen pressure resulted in a larger permittivity

over the entire frequency range. For example, the dielectric constant reached -158 at 10³ Hz for the 4.0 mTorr sample, in contrast with -21 at 1.8 mTorr. The frequency dependence of the dielectric dispersion is highlighted by two distinguishable characteristics, i.e., the abrupt increases in $\varepsilon_r$ toward the lower frequency in the frequency range below -10⁴ Hz and the dipolar resonance peaks at -10⁵ Hz (as highlighted with a red box in Fig. 3g). These characteristics were driven by two polarization mechanisms having different dipole moments at specific frequencies: interfacial (or space-charge) polarization at frequencies below 10⁴ Hz and dipolar (or ferroelectric) polarization in the frequency range of -10⁴ to -10⁵ Hz, as guided with the horizontal lines in Fig. 3g. Note that polycrystalline CCTO is known to possess intrinsic interfacial polarization mainly across grain boundaries, which is driven by defects including oxygen vacancies and resultant defect dipoles, as reported elsewhere[30,31,41,47]. This large contribution by the interfacial polarization is meaningful when considering no grain boundaries present in the amorphous state. The dielectric resonance at -10⁵ Hz became clearer in the case of 4.0 mTorr, which must be related to the stronger ferroelectric nature of the films prepared with a higher $pO_2$, although the contribution of interfacial polarization to the relative permittivity is larger than that of dipolar polarization when the frequency becomes lower. In the 4.0 mTorr sample, for example, the dipolar polarization contributed up to -75 to the permittivity value (as obtained by following the horizontal line to the y-axis in Fig. 3g), and the substantial increase below -10⁴ Hz was enabled by the interfacial polarization.

Figure 3h shows the complex impedance plots of the CCTO thin films deposited at different $pO_2$. The 3.0 and 4.0 mTorr films exhibited

clear semicircles with near-zero intercepts on the Z′ axis at high frequency, whereas the films deposited at 1.8 and 2.3 mTorr present very large semicircles beyond the measurable frequency. Interestingly, the 3.0 and 4.0 mTorr films demonstrated additional small arcs, as shown in the inset of Fig. 3h and reported previously[32]. The two arcs in the amorphous state create two intercepts on the Z′ axis (x-axis); the first and second intercepts indicate the grain boundary resistance and the resistance at the interface between the film and electrode, respectively[32,49]. A higher $pO_2$ decreased the interfacial resistance, which was related to the changed chemical states with fewer oxygen vacancies. As evidence of grain boundaries, the first intercept is still interesting, probably suggesting that a certain irregularity exists that acts similarly as a potential barrier for charge mobilization in local areas[32]. The first and second intercept values of the 4.0 mTorr samples are ~6.1 and ~82.6 kΩ, respectively, where the first resistance is close to the reported grain boundary resistance of 9.3 kΩ for polycrystalline $CaCu_3Ti_4O_{12}$ films[27,32]. Accordingly, a sort of short-range grain boundaries may be present in the amorphous state only at high $pO_2$ (having nearly CCTO stoichiometry) and may, therefore, contribute to the additional interfacial polarization.

Concerning the mechanism of piezoelectric power generation, the local dipoles of $TiO_6$ distributed with no long-range order are believed to be the main reason for the occurrence of electromechanical conversion, basically identical to the case of crystalline CCTO where the $TiO_6$ octahedra are responsible for piezoelectricity. The RN-LBU theory proposed in other amorphous perovskite oxide films[34–36] is also believed to be applicable to this amorphous CCTO film. As observed in the XPS analysis in Fig. 1c, the defect dipoles of $Ti'_{Ti}$ - $V_o^{··}$ modifies the nature of $TiO_6$ octahedra because the oxygen vacancy and reduced $Ti^{3+}$ substitute the lattice oxygen and $Ti^{4+}$ in the octahedra. As another defect dipole of $Cu'_{Cu}$ - $V_o^{··}$ between the reduced $Cu^{1+}$ and oxygen vacancy is assumed to form in the amorphous structure. These defects are commonly recognized in the crystalline CCTO, which are responsible for the interfacial polarization across the grain boundaries[55–57]. Identical defect dipoles were reported as being responsible for the enhanced permittivity in polycrystalline CCTO thin films processed by other deposition techniques, including solution deposition[58] and pulsed laser deposition[59,60]. The involvements of the defect dipoles, which are within the octahedral and connecting the octahedra, are likely the additional contributors to the enhanced polarization because the bending operation extends the lengths of defect dipoles.

At the same time, however, the enhanced polarization increases the dielectric constant and can act negatively to the piezoelectric power generation (because the higher dielectric constant reduces the piezoelectric voltage coefficient). However, the raised dielectric constant competes with the enhanced piezoelectricity for the higher figure of merit (FOM) as a parameter representing the effectiveness of the power generator. The higher piezoelectric coefficient with a lower dielectric constant is demanded for the higher FOM. FOM given by $d_{33} \times g_{33}$, where $g_{33}$ is the piezoelectric voltage coefficient, was estimated after evaluating $g_{33}$ with the relation of $g_{33} = d_{33}/\varepsilon_r\varepsilon_0$ where $\varepsilon_0$ is the zero permittivity[17,61]. The estimated results are presented in Supplementary Table 3. As expected, the sample processed at 4.0 mTorr (that exhibited the maximum harvesting performance) showed the best FOM of $1.22 \times 10^{-12}$ m$^2$ N$^{-1}$, which was substantially raised compared to $0.06 \times 10^{-12}$ m$^2$ N$^{-1}$ at 1.8 mTorr.

### Significance of the best harvesting performance

Next, we applied electric fields to potentially induce extra polarization to further enhance the piezoelectric power generation, as observed in the harvesting performance for the ~497-nm CCTO films in Fig. 4a, b. Applying an electric field up to 120 kV cm$^{-1}$ improved the output, reaching maximum peak values of ~38.7 V and ~1.24 µA (Fig. 4c). The potential mechanism of the poling effect is schematically illustrated in

Fig. 4d, where the planar view of amorphous CCTO structure is presented with disturbed long-range order of perovskite lattices, particularly for the case of 4.0 mTorr, which demonstrated the nearly stoichiometric ratios (see Supplementary Fig. 13 for the unit cell of CCTO crystal). In the amorphous structure, randomly distributed $TiO_6$ octahedra are visualized with two potential defect dipoles of $Ti'_{Ti}$ - $V_o^{··}$ and $Cu'_{Cu}$ - $V_o^{··}$ as described previously. The applied poling field is assumed to contribute to the alignment of the permanent dipoles and the extension of the defect dipoles along the direction of the electric field, resulting in enhanced polarization. Figure 4e shows the dependence of the harvesting performance on the load resistance of the ~497-nm CCTO thin films processed at 4.0 mTorr in the resistance range of ~10$^2$ to ~10$^8$ Ω, which was measured under the optimal bending conditions. As expected, the voltage value gradually rose with the applied load resistance, but the power peaked at ~413 µW at a load resistance of ~10$^6$ Ω. The reliability of the optimal harvester is confirmed by ensuring consistent harvesting performance up to 11,000 cycles, as seen in Supplementary Fig. 14. It is worth mentioning that our harvesting performance of 22.0 V and 906 nA is still viable even without the poling process. There is no such noticeable performance for piezoelectric thin films, particularly without applying the poling field.

The best harvesting outcomes, ~38.7 V and ~413 µW, with the resulting power density of $2.8 \times 10^6$ µW cm$^{-3}$ were projected onto a chart comparing our values with the results reported for typical piezoelectric thin-film-based energy harvesters, as shown in Fig. 4f. All corresponding values are listed in Supplementary Table 4, along with information on the material, film thickness, substrate, and deposition parameters, and measurement conditions. The reported harvesting characteristics were based on thin-film harvesters of various piezoelectric materials, including $Pb(Zr,Ti)O_3$[3,10,11,62], $(K,Na)NbO_3$[63,64], $(Bi,Na)TiO_3$[8], $ZnO$[65], $AlN$[66,67], and perovskite halides[68–71], which were characterized by different measurement techniques utilizing vibration, pressing, and bending. In the case of thin films on rigid substrates, the vibrational operation is used for the cantilevers with the optimized effect at the specific resonant frequency, while the flexible harvesters on polymer substrates utilize the mechanical input sources typically from pressing or bending. To the best of our knowledge, our optimized output power and power density are higher than any values previously reported for piezoelectric thin-film harvesters, even with our relatively thin thickness of ~497 nm and its amorphous nature. As the best thin-film harvester thus far, the thin-film cantilever of 2.8-µm-thick $Pb(Zr,Ti)O_3$ prepared by sputtering delivered 244 µW and $1.1 \times 10^6$ µW cm$^{-3}$ as a result of mechanical vibration at a resonant frequency of 50 Hz[3]. Another noticeable generation result of 182 µW and $2.7 \times 10^5$ µW cm$^{-3}$ by bending was reported for flexible 486-nm-thick methylammonium lead iodide (MAPbI$_3$) films on an ITO-PEN substrate[70]. In addition, our best harvesting performance was compared to the power-generation outcomes reported for polymer-matrix composite harvesters where piezoelectric materials such as PZT, BaTiO$_3$, and perovskite halides are incorporated as fillers in various polymer matrices as listed in Supplementary Table 5[72–86]. Our best values of 413 µW and $2.8 \times 10^6$ µW cm$^{-3}$ are still better than the values achieved for piezoelectric composites even though some of the composite harvesters demonstrated noteworthy output voltage and current values greater than our values (see Supplementary Fig. 15 for each comparison)[80,83–85]. The highest power density value of $4.3 \times 10^4$ µW cm$^{-3}$ was reported for the MASnI$_3$- polyvinylidene fluoride (PVDF) composites[81], which is two orders lower than our best value.

## Discussion

We successfully demonstrated high-performance piezoelectric energy harvesters based on amorphous perovskite CCTO thin films deposited directly onto a plastic substrate without the complicated transfer process usually required for crystalline piezoelectric oxide films. The electromechanical energy conversion was noticeably enhanced with a

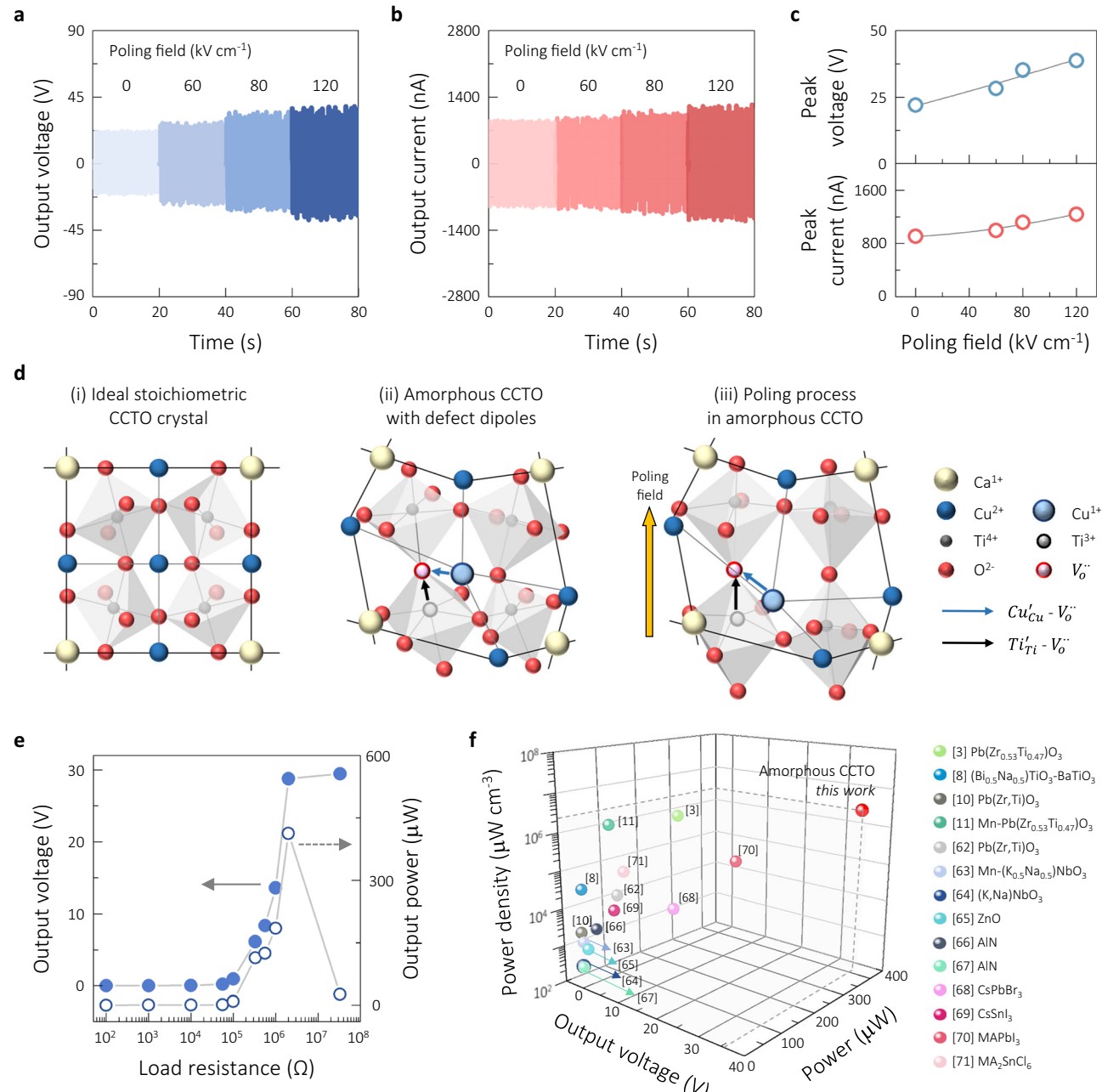

**Fig. 4 | Electric-field-driven harvesting performance and comparative charts.**
**a**, **b** Output voltage and current obtained with the ~497-nm-thick CCTO harvester after applying the electric fields, which were measured under the optimal conditions of 0.77% and 3.10 Hz. **c** Plots of peak voltage and current with increasing poling field. **d** Schematic illustration of CCTO structures: (i) ideal stoichiometry crystal structure with spatially distributed $TiO_6$ octahedra, (ii) amorphous structure in distorted lattices with representative defect dipoles of $Ti'_{Ti}$-$V_o^{..}$ and $Cu'_{Cu}$-$V_o^{..}$, and (iii) effect of poling with extensions of octahedral dipoles and defect dipoles along

with the vertical direction of poling field. **e** Output voltage and power were measured with the increasing load resistance for the ~497-nm-thick CCTO film deposited at 4.0 mTorr, which were measured under the optimal conditions. **f** Plot comparing our best values of voltage, power, and power density with the reported values for thin-film-based piezoelectric energy harvesters for various piezoelectric materials, including ZnO, AlN, PZT, perovskite halides. Detailed information on the values cited in the plot is listed in Supplementary Table 4. Source data are provided as a Source data file.

higher oxygen partial pressure and thicker film. Piezoelectric characteristics of the amorphous CCTO films are assumed to originate mainly from dipolar polarization driven by local $TiO_6$ octahedra ordered in short range, which differs from regular octahedral networks connected via apex-to-apex in the crystalline state. As another contributor, defect dipoles depending on $pO_2$ create additional interfacial polarization. Piezoelectric power generation is likely associated with the competing effect between the enhanced piezoelectricity and relative permittivity, which are driven by both dipolar polarization and

interfacial polarization. As optimal harvesting outcomes with poling, an output voltage of 38.7 V, power of 413 μW, and power density of $2.8 \times 10^6$ μW cm$^{-3}$ were attained for nearly stoichiometric amorphous thin films processed with the highest oxygen pressure. The best harvesting performance far exceeds the values reported for any energy harvesters based on typical perovskite oxide thin-film cantilevers mostly relying on the vibrational motion. Presumably, the bending operation may also produce a higher input force for higher electromechanical conversion compared to the vibrational mode.

## Methods

### Deposition of amorphous CCTO thin films

CCTO thin films were deposited at room temperature on PEN (~125 μm thick) substrates coated with ~100 nm of platinum by RF magnetron sputtering using a 2-inch stoichiometric CCTO target. The CCTO target was prepared by a conventional solid-state reaction at 1000 °C and then hot-pressing at 1100 °C under a pressure of 15 MPa. Although the distance between the target and substrate was kept at 7 cm, deposition was carried out at a constant working pressure of 7.2 mTorr and an RF power of 100 W by using different argon–oxygen gas mixtures. The different gas mixtures are expressed in terms of the oxygen partial pressure from 1.8 to 4.0 mTorr. For example, a pressure of 4.0 mTorr was obtained from an $Ar/O_2$ ratio of 30/38 in sccm. Samples with thicknesses ranging from ~86 to ~497 nm were prepared by extending the sputtering time for up to ~177 min. The substrate holder was slowly rotated at eight rotations per min about its axis during film deposition.

### Preparation of piezoelectric energy harvesters

The piezoelectric energy harvesters were fabricated by covering the CCTO film on Pt/PEN with another PDMS-coated ITO/PEN substrate to complete the harvester structure of PEN/ITO/PDMS/CCTO/Pt/PEN with an effective area of $3 \, cm^2$. To prepare the PDMS-coated ITO/PEN substrate, a PDMS (Sylgard 184, Dow Corning, USA) solution containing the base monomer with 10 wt% curing agent was spin-coated onto a 50-μm-thick ITO/PEN substrate at 4000 rpm for 60 s and pre-cured at 120 °C for 5 min. The pre-cured PDMS layer was adhered to the CCTO film by curing at 120 °C for a longer duration of 30 min. Cu wires were externally connected to the exposed Pt prior to passivating both sides of the harvester using polyimide (PI) tape.

### Characterization and measurement

The crystal structure of the deposited CCTO films was examined using XRD (SmartLab, Rigaku, Japan) in the $2\theta$ range of 30–80°. High-resolution TEM (JEM-ARM200F, JEOL, Japan) equipped with a high-angle annular dark field (HAADF) detector was used to observe the detailed microstructures of the specimens prepared using a standard focused ion beam (FIB) technique. The chemical states of the films were analyzed by high-resolution XPS (VG ESCALAB 220i-XL, Thermo Fisher Scientific, USA) with Al-Kα photons (1486.6 eV) at a base pressure of $1.1 \times 10^{-10}$ Torr in an ultrahigh vacuum chamber. The XPS spectra were calibrated to the C 1s peak at 284.8 eV. The dielectric constant was measured in the frequency range of $10^2$–$10^6$ Hz using an impedance analyzer (4294 A, Agilent, USA). The piezoelectricity of the CCTO films was assessed by PFM (Multimode 8, Bruker, Germany) using a conductive Pt/Ir-coated Si cantilever tip (PPP-NCHPt, Nanosensors, USA) having a spring constant of $42 \, N \, m^{-1}$. The PR amplitude and phase images were obtained by scanning the film surface at 40 kHz under an AC bias of 4 V in lock-in mode. The effect of DC bias on the amplitudes was examined in the range of −10 V and +10 V. The effective piezoelectric coefficient $d_{33,eff}$ was calculated from the peak amplitudes at +10 V in the amplitude-DC bias curves. Specially, the mapping images of $d_{33,eff}$ were constructed from 400 data points over an extended area of $10 \times 10 \, \mu m^2$. Ferroelectric domain reversal was verified by applying alternate DC fields of −10 V and + 10 V to the designated areas. Electric field-dependent displacement was measured using an aixACCT TF analyzer 2000E (aixAcct Systems GmbH, Germany) equipped with a single-beam laser interferometer in the condition of ±3 V triangular excitation at 10 Hz for the films deposited on Si. From the displacement data, the effective piezoelectric coefficient was estimated by the linear regression from point $N/4$ to below $N/2$ at the positive side, where $N$ is the total number of measurement points. In addition, the effective transverse piezoelectric coefficient $-e_{31,f}$ was measured at 1 Hz using a four-point bending measurement unit (aix 4PB, aixAcct Systems GmbH, Germany) in combination with the TF analyzer for the films on Si.

The energy-harvesting performance was evaluated under periodic bending using a one-axis high-speed fatigue machine (CTLM500, Ceratorq Inc., Republic of Korea) while varying the bending frequency (0.86–3.50 Hz) and bending strain (0.45–0.77%). The output voltage was measured using a nanovoltmeter (Keithley 2182A, ValueTronics, USA) operating at an internal resistance of 10 MΩ, and the output current was obtained using a galvanostat system (IviumStat, Ivium Technologies, The Netherlands) operating at 1 MΩ. To estimate the power density, the output voltage was measured by changing the load resistance from ~$10^2$ to ~$10^8$ Ω.

## Data availability

The data that support the findings of this work are available from the corresponding author upon request. Source data are provided with this paper.

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

## Acknowledgements

This work was financially supported by grants from the National Research Foundation of Korea (NRF-2021R1A2C2013501) and the Creative Materials Discovery Program of the Ministry of Science and ICT (2018M3D1A1058536).

## Author contributions

J.H. and S.H.P. contributed equally to this work. J.H. and S.H.P. prepared samples and conducted dielectric and piezoelectric measurements. Y.S.J. supported the sample preparation and characterization. J.H. and S.H.P. wrote the main part of the paper. Y.S.C. supervised this study and completed the paper.

## Competing interests

The authors declare no competing interests.
