## [Peer Review File · Nature Communications]

High-Performance Piezoelectric Energy Harvesting in Amorphous Perovskite Thin Films Deposited Directly on a Plastic SubstrateREVIEWER COMMENTS

Reviewer #1 (Remarks to the Author):

Overall comments: This manuscript described flexible piezoelectric energy harvesters based on amorphous $\text{CaCu}_3\text{Ti}_4\text{O}_{12}$ (CCTO) thin film directly deposited onto a plastic substrate without any transfer and high-temperature processes for improving the crystallinity of piezo-ceramics. The fabricated device generates ~ 35.5 V and ~ 1.1 μA of high output signals under bending conditions although thin active layer because of the defect dipoles at interfaces. The piezoelectric properties of the amorphous CCTO films were also thoroughly characterized using PFM techniques. I think that this work is interesting and the authors have provided a systematic study on the topic without critical defects. However, there are some concerns related to materials and device properties that need to be addressed in order to improve the quality of this manuscript. Therefore, I recommend this manuscript is acceptable to Nature Communications after addressing the following major comments.

Comment 1: The reviewer suggests that the authors include the following appropriate articles in the introduction part of the manuscript to support the background of this research more effectively:

- A Reconfigurable Rectified Flexible Energy Harvester via Solid-State Single Crystal Grown PMN–PZT, *Adv. Energy Mater.* 5, 1500051, 2015
- Flexible highly-effective energy harvester via crystallographic and computational control of nanointerfacial morphotropic piezoelectric thin film, *Nano Res.* 10, 437, 2017
- Biomimetic and flexible piezoelectric mobile acoustic sensors with multiresonant ultrathin structures for machine learning biometrics, *Sci. Adv.* 7, eabe5683, 2021
- Basilar membrane-inspired self-powered acoustic sensor enabled by highly sensitive multi tunable frequency band, *Nano Energy* 53, 198, 2018

Comment 2: Regarding the device structure, it seems that the PDMS dielectric layer is significantly thicker than the CCTO layer, which could negatively impact the device's output performance and potentially cause triboelectric effects between the PDMS and CCTO during bending. Is there a specific reason why the top electrode was not directly deposited onto the CCTO layer? Additionally, was the attachment between the PDMS and piezoelectric layer stable after 3600 bending cycles?

Comment 3: It is important to provide not only the d_{33} but also the piezoelectric voltage constant (g_{33}) and the piezoelectric figure of merit (FoM), as they are important indicators of the energy harvester's performance. The authors should include these values in the manuscript.

Comment 4: On page 12, the authors stated that "The effect of electric field on the defect-driven polarization has not been common: ~". It is recommended that the authors provide a scheme showing the difference between poling in the common case and the defect-driven case to help readers understand.

Comment 5: In Supplementary Table 1, the authors compared the results of cantilever-type devices measured under vibration conditions with the results of this study, but there are significant differences in the device structure and measurement method used in the manuscript. Therefore, the comparison group seems inappropriate and should be appropriately revised unless there is a specific reason.

Comment 6: In the Methods section, the authors attached Cu wires to the electrodes using PI tape, which may not provide good electrical/mechanical connectivity. It is recommended to fabricate a new device using commonly used conductive epoxy or other more appropriate methods for wire bonding.

Comment 7: The symbol ϵ is used as an abbreviation for dielectric constant and bending strain, simultaneously. They should be replaced with different symbols to reduce reader confusion.

Reviewer #2 (Remarks to the Author):

The author has studied the energy harvesting properties of amorphous thin films based on perovskite $\text{CaCu}_3\text{Ti}_4\text{O}_{12}$ (CCTO) thin films employing a magnetron sputtering process over a flexible substrate. The device fabrication seems defective and the energy harvesting studies are insufficient to recommend this manuscript for publication. The following comments are to improve the quality of the manuscript.

Comments:

1. The author has used amorphous $\text{CaCu}_3\text{Ti}_4\text{O}_{12}$ (CCTO) for energy harvesting applications. Piezoelectricity is an internal property of the material which depends on the crystal structure and orientation of lattice planes. However, the mechanism of charge generation in the amorphous CCTO structure is unclear and should be explored scientifically.
2. The novelty of employing CCTO for harvesting energy is not justified. Various piezoceramics and polymer composites with piezo active fillers perform better with respect to the reported performance of the CCTO. So, the author is suggested to explain the advantages of reporting method and material as compared to other piezoelectric materials.
3. How to confirm the film stoichiometry is exactly equal to $\text{CaCu}_3\text{Ti}_4\text{O}_{12}$? Why not the formation of another compound? Non-centrosymmetric nature is the basic requirement for producing the

piezoelectric effect in any compound. In the present case, how amorphous CCTO is producing piezoelectricity, and how does it differ from the crystalline CCTO film?

4. The author has followed the direct growth of CCTO over ITO-coated PEN film. What is the significance of the PDMS layer over the active piezo film? The device fabrication seems defective, and the output may be influenced by PDMS triboelectrification. So, the device needs to be refabricated with the proper precaution to measure the inherent performance under the external stimulus.

5. PFM characterization is adopted to demonstrate the piezoelectric properties of CCTO film, but it is a localized concept and may not be considered for direct evidence. It should be considered for the supporting evidence along with the standard piezoelectric measurements. Moreover, the PFM results are easily influenced by electrical interference. Therefore, authors should provide piezoelectric coefficient values of CCTO film by DBLI.

6. Author reported that the d_{33} , eff values of CCTO film increased from 2.1 pm/V to 35.3 pm/V. What is the reason behind such drastic improvement of d_{33} , eff on varying the partial oxygen pressure?

7. The PENG device portrays some noisy peaks in the output voltage and current. The peak-to-peak output gradually increases but stable performance enhancement is not observed in the films grown at various oxygen partial pressure.

8. The stability is carried out at a single frequency. The author needs to discuss the impact of various frequencies over the piezo film or needs to provide the frequency optimization for the enhanced output of the PENG device.

9. The author has followed electrode poling for 60-80 kV/cm to optimize the performance of the PENG device. The electrode poling of 80kV/cm does not provide any significant performance improvement even after the application of very high voltage.

10. The overall physicochemical and energy harvesting performance is inadequate to confirm the origin of piezoelectricity in amorphous CCTO film.

Reviewer #3 (Remarks to the Author):

This work developed a flexible amorphous thin-film energy harvester based on perovskite $\text{CaCu}_3\text{Ti}_4\text{O}_{12}$ (CCTO) thin films on a plastic substrate. They attributed the high-performance output of this piezoelectric nanogenerator to the unusually high permittivity of amorphous CCTO film, which was caused by interfacial and dipolar polarization mechanisms depending on the defect states and chemical migration under the bending operation. Evidences of XPS, PFM, impedance, and dielectric measurements were also provided to confirm their hypothesis.

However, there are some important questions to be clarified :

1. According to the authors' claim, the resultant amorphous nature of the films resulted in an output voltage and current of ~ 35.5 V and ~ 1.1 μA , respectively, which tops the previously reported record for typical polycrystalline ferroelectric oxide thin-film cantilevers. What are the advantages of this perovskite oxide compared to halide perovskites? Compared to this CCTO thin film, halide perovskite film can be prepared by a solution-based method at room temperature (suited for flexible substrates), which is much easier and more convenient than RF magnetron sputtering in this work. On the other hand, some halide perovskites have shown better output performance than this work. For example, Jella et al. reported a MAPbI_3 -PVDF composite-based PENG with higher outputs of 45.6 V and 4.7 $\mu\text{A}/\text{cm}^2$ (Nano Energy 2018, 53, 46–56); Khan et al. reported a porous FAPbBr_2 -PVDF composite-based PENG with higher outputs of 85 V and 30 μA (J. Mater. Chem. A, 2020, 8, 13619–13629) and a ferroelectric $(\text{ATHP})_2\text{PbBr}_2\text{Cl}_2$ -PVDF composite with high outputs of 90V and 6.5 μA (Nano Energy 2021, 86, 106039); Huang et al. also designed a ferroelectric halide perovskite $\text{TlMg}_2\text{SnCl}_6$ with a large d_{33} of 137 pC/N and g_{33} of 0.98 V·m/N, exhibiting high output performance of 81 V and 2 μA (ACS Energy Lett. 2021, 6, 16–23). All these halide perovskites show both great output performance and flexibility.

2. The author claimed that oxygen partial pressure $p\text{O}_2$ was a very important parameter that affected the harvesting performance of the device. How about the stability of this device in an ambient environment? What is the durability of the performance that can be maintained at ambient condition? In addition, whether the performance will be affected if the device was put into a high-vacuum condition.

3. Would the ambient humidity affect the stability and performance of this device? Evaluation as a function of humidity would be required.

4. As for Figure 3d, the authors claimed that the dipolar polarization contributed up to ~ 75 to the permittivity value for 4.0 mTorr sample, which was determined by following the horizontal line to the y-axis in Fig.3d. From this figure, we can also notice that a sharp increase and decrease of dielectric constant existing at the ~ 105 frequency. The authors should provide more details to explain why the dielectric constant change in this manner.

How does the dipolar polarization affect the shape of this dielectric curve? Are there any different interactions existing in this frequency?

Response to the reviewers' comments

Nature Communications manuscript: NCOMMS-23-11198

TITLE: High-Performance Piezoelectric Energy Harvesting in Amorphous Perovskite Thin Films Deposited Directly on a Plastic Substrate

Reviewer #1:

Overall comments: This manuscript described flexible piezoelectric energy harvesters based on amorphous $\text{CaCu}_3\text{Ti}_4\text{O}_{12}$ (CCTO) thin film directly deposited onto a plastic substrate without any transfer and high-temperature processes for improving the crystallinity of piezo-ceramics. The fabricated device generates ~ 35.5 V and ~ 1.1 μA of high output signals under bending conditions although thin active layer because of the defect dipoles at interfaces. The piezoelectric properties of the amorphous CCTO films were also thoroughly characterized using PFM techniques. I think that this work is interesting and the authors have provided a systematic study on the topic without critical defects. However, there are some concerns related to materials and device properties that need to be addressed in order to improve the quality of this manuscript. Therefore, I recommend this manuscript is acceptable to Nature Communications after addressing the following major comments.

Comment 1: The reviewer suggests that the authors include the following appropriate articles in the introduction part of the manuscript to support the background of this research more effectively:

- A Reconfigurable Rectified Flexible Energy Harvester via Solid-State Single Crystal Grown PMN–PZT, *Adv. Energy Mater.* 5, 1500051, 2015
- Flexible highly-effective energy harvester via crystallographic and computational control of nanointerfacial morphotropic piezoelectric thin film, *Nano Res.* 10, 437, 2017
- Biomimetic and flexible piezoelectric mobile acoustic sensors with multiresonant ultrathin structures for machine learning biometrics, *Sci. Adv.* 7, eabe5683, 2021
- Basilar membrane-inspired self-powered acoustic sensor enabled by highly sensitive multi tunable frequency band, *Nano Energy* 53, 198, 2018

REPLY) The suggested articles were included in the Introduction, as examples of using piezoelectricity to convert mechanical input into either electrical signal or power for sensing and energy-harvesting applications. The corresponding references of [4,5,15,16] were newly added.

Comment 2: Regarding the device structure, it seems that the PDMS dielectric layer is significantly thicker than the CCTO layer, which could negatively impact the device's output performance and potentially cause triboelectric effects between the PDMS and CCTO during bending. Is there a specific reason why the top electrode was not directly deposited onto the CCTO layer? Additionally, was the attachment between the PDMS and piezoelectric layer stable after 3600 bending cycles?

REPLY) We agree that the potential gap between the PDMS and CCTO layers may create the triboelectric effect. The PDMS layer was used to provide more stable harvesting performance with consistent outcomes over a long period of time. Note that we carefully applied two-step processing for the intimate adhesion of the PDMS layer: the pre-curing of the spin-coated PDMS layer at 120 °C

for 5 min and then annealing with the contacted CCTO layer at 120 °C for 30 min. We should mention that similar energy harvesting outcomes were produced even without the PDMS layer, but the consistency of peak-to-peak values was greatly improved with the PDMS layer.

The following harvesting performance compares the harvesting outcomes with and without using the PDMS layer. Although the output voltage and current values are similar between the two cases, more stable harvesting performance is evident with the PDMS layer. The result with the PDMS layer was added as Supplementary Fig. 2.

New Supplementary Fig. 2. Comparison of output performance, **a** voltage and **b** current, of the harvesters with and without the intermediate PDMS layer.

We also confirmed that only the PDMS layer (without the active CCTO layer) did not produce significant energy harvesting performance after the double-curing process as seen below, indicating that the energy harvesting outcomes were generated dominantly by the CCTO layer. The result was included as Supplementary Fig. 3.

Supplementary Fig. 3. **a** Output voltage and **b** output current measured with only the PDMS layer (i.e., without the CCTO film), indicating the trivial contributions of the PDMS layer.

As suggested, we extended the bending cycle up to 11,000 cycles (beyond the previous 3,600 cycles) to ascertain the stable harvesting performance. The noticeable stability was observed over the extended cycles, indicating the well-maintained attachment between the PDMS and CCTO layers. Otherwise, the outcome may be deteriorated with the damaged mechanical integrity (and thus poor electrical connectivity) in the device structure or may be interrupted presumably owing to triboelectricity if the gap between the PDMS and CCTO layers is created during the extended cycling test.

Revised Fig. 2f. Stability evaluation of the output voltage over 11,000 cycles for the CCTO thin film deposited at 4.0 mTorr, measured at the bending strain of 0.77 % and the bending frequency of 2.73 Hz.

In this regard, the following paragraph was added:

“It should be mentioned that the intermediate PDMS layer was used to generate more stable energy harvesting performance as seen in Supplementary Fig. 2 where more consistent peak-to-peak output values were created with the incorporation of PDMS layer compared to the results produced by the CCTO layer without PDMS. The harvesting performance of the PDMS layer alone was trivial, as demonstrated in Supplementary Fig. 3, as it generated an output voltage and current of only ~ 0.05 V and ~ 17 nA, respectively”

(in page 7, line 19)

Comment 3: It is important to provide not only the d_{33} but also the piezoelectric voltage constant (g_{33}) and the piezoelectric figure of merit (FoM), as they are important indicators of the energy harvester's performance. The authors should include these values in the manuscript.

REPLY) As suggested, we provided the piezoelectric voltage coefficient g_{33} (given by $g_{33} = d_{33}/\epsilon_r\epsilon_0$ where ϵ_r is the relative permittivity and ϵ_0 is the zero permittivity) and the figure of merit (FOM: given by $d_{33} \times g_{33}$) as extra parameters concerning the harvesting performance. Relative permittivity of each sample was extracted from Fig. 3g where the dielectric dispersion is shown as a function of frequency. The ϵ_r value at 40 kHz was selected because d_{33} was obtained from the PFM measurement operated under the AC field of this frequency. As expected, the sample processed at 4.0 mTorr (which exhibited the maximum harvesting performance) showed the best FOM of $1.22 \times 10^{-12} \text{ m}^2 \text{ N}^{-1}$, which was substantially raised compared to $0.06 \times 10^{-12} \text{ m}^2 \text{ N}^{-1}$ at 1.8 mTorr, as listed below. The estimated results were included as new Supplementary Table 2.

Supplementary Table 2. Estimation of the piezoelectric voltage coefficient g_{33} and the figure of merit (FOM) for the harvesters based on amorphous CCTO thin films processed at different oxygen partial pressure. Note that we used the dielectric constant values measured at 40 kHz to match with the PFM frequency.

Oxygen partial pressure	d_{33} (pm V ⁻¹)	ϵ_r (ϵ_{33}/ϵ_0)	g_{33} (m V N ⁻¹)	FOM ($10^{-12} \text{ m}^2 \text{ N}^{-1}$)
1.8 mTorr	1.73	5	0.039	0.06
2.3 mTorr	5.16	12	0.049	0.25
3.0 mTorr	17.8	38	0.052	0.94
4.0 mTorr	27.7	71	0.044	1.22

In this regard, the following description was added in text:

“The higher piezoelectric coefficient with a lower dielectric constant is demanded for the higher FOM. FOM given by $d_{33} \times g_{33}$, where g_{33} is the piezoelectric voltage coefficient, was estimated after evaluating g_{33} with the relation of $g_{33} = d_{33}/\epsilon_r \epsilon_0$ where ϵ_0 is the zero permittivity^{17,56}. The estimated results are presented in Supplementary Table 2. As expected, the sample processed at 4.0 mTorr (that exhibited the maximum harvesting performance) showed the best FOM of $1.22 \times 10^{-12} \text{ m}^2 \text{ N}^{-1}$, which was substantially raised compared to $0.06 \times 10^{-12} \text{ m}^2 \text{ N}^{-1}$ at 1.8 mTorr.”

(in page 14, line 13)

Comment 4: On page 12, the authors stated that "The effect of electric field on the defect-driven polarization has not been common: ~". It is recommended that the authors provide a scheme showing the difference between poling in the common case and the defect-driven case to help readers understand.

REPLY) As suggested, we added the schematic of the poling process concerning the creation of extra polarization in amorphous CCTO thin films. As seen below, the schematic consists of three illustrations of CCTO structure: (i) typical crystalline CCTO structure reflecting ideal stoichiometry with the spatial distribution of TiO_6 octahedra, (ii) amorphous structure of CCTO with distorted lattices and thus destroyed long-range atomic arrangements with examples of defect dipoles (Note that the optimal 4.0 mTorr sample possessed stoichiometry CCTO ratios even in the amorphous state according to the XPS result in Fig. 1c and Supplementary Table 1), and (iii) the effects of poling, which were highlighted with the arrangements of the octahedra and the extension of defect dipoles, both along with the direction of applied electric field.

New Fig. 4d. Schematic illustration of CCTO structures: (i) ideal stoichiometry crystal structure with spatially distributed TiO_6 octahedra, (ii) amorphous structure in distorted lattices with representative defect dipoles of $\text{Ti}'_{\text{Ti}} - \text{V}_o^{\bullet\bullet}$ and $\text{Cu}'_{\text{Cu}} - \text{V}_o^{\bullet\bullet}$, and (iii) effect of poling with extensions of octahedral dipoles and defect dipoles along with the vertical direction of poling field.

The schematic was added as new Fig. 4d with the following description:

“The potential mechanism on the poling effect is schematically illustrated in Fig. 4d where the planar view of amorphous CCTO structure is presented with disturbed long-range order of perovskite lattices, particularly for the case of 4.0 mTorr which demonstrated the nearly stoichiometric ratios (see Supplementary Fig. 13 for the unit cell of CCTO crystal). In the amorphous structure, randomly distributed TiO_6 octahedra are visualized with two potential defect dipoles of $\text{Ti}'_{\text{Ti}} - \text{V}_o^{\bullet\bullet}$ and $\text{Cu}'_{\text{Cu}} - \text{V}_o^{\bullet\bullet}$ ”

V_o'' . The applied poling field is assumed to contribute to the alignment of the permanent dipoles and the extension of the defect-dipoles along the direction of electric field, resulting in the enhanced polarization.”

(in page 16, line 1)

NOTE: The crystal schematic above was based on the plane view of the following 3D structure of stoichiometric CCTO unit cells (which was included as Supplementary Fig. 13)

New Supplementary Fig. 13. Crystal structure of stoichiometry CCTO unit cell

Comment 5: In Supplementary Table 1, the authors compared the results of cantilever-type devices measured under vibration conditions with the results of this study, but there are significant differences in the device structure and measurement method used in the manuscript. Therefore, the comparison group seems inappropriate and should be appropriately revised unless there is a specific reason.

REPLY) As suggested, we agree that the harvesting performance is very hard to be simply compared with the reported output values because the harvesting results depend on sample dimension, device structure (with electrode pattern), and measurement technique and condition. For example, multi-stacking layered structure or interdigitated electrode creates much higher current values, while the thicker films produce the higher voltages in the same condition. As typically reported, we added a column of power density (per volume) in a revised Table for more reasonable comparison even though there can be still misreading on the performance comparison. In fact, we combined two previous comparative Supplementary Table 1 (about the thin-film cantilevers based on perovskite oxides) and Supplementary Table 2 (flexible harvesters) into one Table. The revised Supplementary Table S3 can be seen below, in which the thin film-based piezoelectric energy harvesters deposited directly on rigid or flexible substrates are listed with the information on material, processing, dimension, measurement technique, and power generation parameters as being cited from each reference. Note that we also included recent reports on perovskite halide-thin-film-based piezoelectric energy harvesters.

Based on the revised Supplementary Table 3, we also made a new plot projecting the values of output voltage, power, and power density per each case as seen below. The plot was added as a revised Fig. 4f. Note that our voltage, power and power density values correspond to the record-values in the field of thin-film-based piezoelectric energy harvesters.

Revised Supplementary Table 3. Comparison of our best harvesting performance with representative results reported for piezoelectric thin-film harvesters based on various piezoelectric materials, such as perovskite oxides, ZnO, AlN, and perovskite halides.

Material (Substrate)	Deposition method	Film thickness (nm)	Poling field (kV cm ⁻¹)	Output voltage (V)	Output current (μA)	Power (μW)	Power density (μW cm ⁻³)	Mechanical input source (condition)	Ref.
Pb(Zr _{0.53} Ti _{0.47})O ₃ (Stainless steel)	Sputtering	2,800	Poled	2.6	N/A	244	1.1 × 10 ⁶	Vibration (50 Hz)	[3]
Pb(Zr,Ti)O ₃ (Si)	Spin-coating	1,000	Poled	0.16	N/A	2.15	3.2 × 10 ³	Vibration (462 Hz)	[10]
Mn-Pb(Zr _{0.52} Ti _{0.48})O ₃ (Ni foil)	Sputtering	3,000	Poled	2.44	N/A	60	1.6 × 10 ⁶	Vibration (72 Hz)	[11]
Mn-(K _{0.5} Na _{0.5})NbO ₃ (Si)	Spin-coating	1,000	120	0.52	N/A	3.6	1.8 × 10 ³	Vibration (132 Hz)	[58]
(K,Na)NbO ₃	Sputtering	2,200	Poled	0.42	N/A	1.6	4.1 × 10 ²	Vibration (393 Hz)	[59]
AlN (Si)	Sputtering	500	No poling	2.48	N/A	20.5	4.1 × 10 ³	Vibration (210 Hz)	[61]
(Bi _{0.5} Na _{0.5})TiO ₃ -BaTiO ₃ (Si)	Spin-coating	2,000	600	0.75	N/A	2.22	4.6 × 10 ⁴	Vibration (42 Hz)	[8]
PZT (Stainless steel)	Sputtering	3,000	Poled	9.4	N/A	13.4	5.3 × 10 ⁴	Pressing (finger)	[57]
ZnO (PET)	Sputtering	2,000	No poling	2.25	N/A	0.28	1.4 × 10 ³	Vibration (370 Hz)	[60]
AlN (PI)	Sputtering	900	No poling	0.7	N/A	1.4 × 10 ⁻³	4.0 × 10 ²	Bending (N/A)	[62]
CsPbBr ₃ (ITO-PEN)	Spin-coating	545	35	22.6	1.13	21.3	5.6 × 10 ⁴	Bending (strain, 0.67%)	[63]
CsSnI ₃ (ITO-PEN)	Spin-coating	354	17.4	9.5	0.45	4.23	2.4 × 10 ⁴	Bending (N/A)	[64]
MAPbI ₃ (ITO-PEN)	Spin-coating	486	11.3	23.1	1.70	182	2.7 × 10 ⁵	Bending (strain, 0.47%)	[65]
MA ₂ SnCl ₆ (ITO-PET)	Spin-coating	300	50	12	1.20	7.33	2.4 × 10 ⁵	Pressing (0.5 MPa)	[66]
Amorphous CaCu ₃ Ti ₄ O ₁₂ (PEN)	Sputtering	497	120	38.7	1.24	413	2.8 × 10 ⁶	Bending (strain, 0.77%)	This work

MA: methylammonium, PI: polyimide, PET: Polyethylene terephthalate

Revised Fig. 4f. Plot comparing our best values of voltage, power, and power density with the reported values for thin-film-based piezoelectric energy harvesters for various piezoelectric materials including ZnO, AlN, PZT, perovskite halides. Detailed information on the values cited in the plot is listed in Supplementary Table 3.

In this regard, we revised the corresponding description as following:

“The best harvesting outcomes, ~ 38.7 V and ~ 413 μW , with the resulting power density of 2.8×10^6 $\mu\text{W cm}^{-3}$ were projected onto a chart comparing our values with the results reported for typical piezoelectric thin-film-based energy harvesters, as shown in Fig. 4f. All corresponding values are listed in Supplementary Table 3, along with information on the material, film thickness, substrate, and deposition parameters, and measurement conditions. The reported harvesting characteristics were based on thin-film harvesters of various piezoelectric materials including $\text{Pb}(\text{Zr,Ti})\text{O}_3$ ^{3,10,11,57}, $(\text{K,Na})\text{NbO}_3$ ^{58,59}, $\text{Bi}(\text{Na,Ti})\text{O}_3$ ⁸, ZnO ⁶⁰, AlN ^{61,62}, and perovskite halides⁶³⁻⁶⁶, which were characterized by different measurement techniques utilizing vibration, pressing, and bending. In the case of thin films on rigid substrates, the vibrational operation is used for the cantilevers with the optimized effect at specific resonant frequency, while the flexible harvesters on polymer substrates utilizes the mechanical input sources typically from pressing or bending. To the best of our knowledge, our optimized output power and power density are higher than any values previously reported for piezoelectric thin-film harvesters, even with our relatively thin thickness of ~ 497 nm and its amorphous nature. As the best thin-film harvester thus far, the thin-film cantilever of 2.8- μm -thick $\text{Pb}(\text{Zr,Ti})\text{O}_3$ prepared by sputtering delivered 244 μW and 1.1×10^6 $\mu\text{W cm}^{-3}$ as a result of mechanical vibration at a resonant frequency of 50 Hz³. Another noticeable generation result of 182 μW and 2.7×10^5 $\mu\text{W cm}^{-3}$ by bending was reported for flexible 486-nm-thick methylammonium lead iodide (MAPbI_3) films on an ITO-PEN substrate⁶⁵.”

(in page 16, line 13)

NOTE: we would like to mention that an additional comparative Supplementary Table 4 comparing our best values with the case of halide-polymer composites (as requested by the third reviewer as will be seen below in this response). Our power density value was still better than the values obtained for any halide-based composite case.

Comment 6: In the Methods section, the authors attached Cu wires to the electrodes using PI tape, which may not provide good electrical/mechanical connectivity. It is recommended to fabricate a new device using commonly used conductive epoxy or other more appropriate methods for wire bonding.

REPLY) As suggested, we made extra harvester devices using a Ag-based conductive epoxy (8331S, MG Chemicals) for the connection of Cu wires (rather than using the previous PI tape). The testing results were basically identical between the usages of the PI tape and conductive epoxy as seen below. Nonetheless, we agree that the conductive epoxy is more generally accepted as a good contact way for proper wire bonding.

Comparison in energy harvesting performance between the usages of PI tape and conductive epoxy for the attachment of Cu wires

Comment 7: The symbol ϵ is used as an abbreviation for dielectric constant and bending strain, simultaneously. They should be replaced with different symbols to reduce reader confusion.

REPLY) As suggested, the symbol of bending strain as changed into ' s_b ' to avoid confusion.

Reviewer #2:

The author has studied the energy harvesting properties of amorphous thin films based on perovskite $\text{CaCu}_3\text{Ti}_4\text{O}_{12}$ (CCTO) thin films employing a magnetron sputtering process over a flexible substrate. The device fabrication seems defective and the energy harvesting studies are insufficient to recommend this manuscript for publication. The following comments are to improve the quality of the manuscript.

Comments:

1. The author has used amorphous $\text{CaCu}_3\text{Ti}_4\text{O}_{12}$ (CCTO) for energy harvesting applications. Piezoelectricity is an internal property of the material which depends on the crystal structure and orientation of lattice planes. However, the mechanism of charge generation in the amorphous CCTO structure is unclear and should be explored scientifically.

REPLY) As you mentioned about the mechanism of charge generation in the amorphous CCTO films, the previous description may not be sufficient for clear explanation. Therefore, we substantially revised the corresponding sections to explain clearly the origin of piezoelectricity and the mechanism of power-generation in the amorphous state, as the first case not reported so far.

It is true that piezoelectricity is an internal property that depends on crystal structure and orientation of lattice planes, particularly from the asymmetric nature of crystals. Because of the amorphous state of CCTO films, crystal structure (with long-range order) does not exist and the regular distortion concerning oxygen octahedra is not expected. Interestingly, however, there have been a few reports on dipolar polarization in cases of amorphous perovskite oxide films, e.g., amorphous BaTiO_3 (*Phys. Rev. B.* **71** (2005) 024116) and SrTiO_3 (*Phys. Rev. Lett.* **99** (2007) 215502 and *Adv. Mater.* **19** (2007) 1515), which experimentally demonstrated that the magnitude of Ti off-centered displacement was higher than the case of crystalline counterparts owing to the weak orientational ordering of TiO_6 octahedra in the amorphous state. Such amorphous state concerning the dipolar polarization was explained based on the ‘random network of local bonding units (RN-LBU) theory’ proposing that the piezoelectricity can exist in structures that lack the spatial periodicity inherent for ionic crystals, but are composed of polar units with directional ordering in short-range scale. However, the overall polarity may not be greatly effective in the disordered structure, partially because the neighboring octahedra can be connected via edges and faces (not via apex-to-apex). In addition, there are rare reports on effective piezoelectric coefficients measured by PFM in the cases of amorphous perovskite oxide films, such as 16 pm V^{-1} for 300-nm-thick KNbO_3 film (*ACS Appl. Mater. Interfaces* **9** (2017) 43220) and $10 (\pm 25\%) \text{ pm V}^{-1}$ for 100-nm-thick SrTiO_3 film (*Adv. Mater.* **19** (2007) 1515), even though the PFM results may not represent intrinsic values. All the cited films above were deposited by sputtering.

Crystalline CCTO is a unique material having intrinsically two polarization mechanisms: dipolar polarization (concerning ferroelectricity) and space-charge (or interfacial) polarization, resulting in extraordinary high dielectric constant $>10^4$ (due to the main contribution by the interfacial polarization with defect dipoles). In this regard, we chased whether the two mechanisms can also be present in the amorphous state with supporting experimental evidence.

Concerning the dipolar polarization (as the origin of ferroelectricity), our amorphous CCTO films deposited at the highest oxygen partial pressure, i.e., 4.0 mTorr (having nearly stoichiometric CCTO

ratios as will be shown here later), seemed to possess local octahedral dipoles as evidenced by the dielectric resonance at 10^5 Hz (in Fig. 3g), the domain switching with reversed DC bias (in Fig. 3e), and the poling effect on harvesting outcomes (in Fig. 4a). Note that crystalline CCTO possesses ferroelectricity as reported (*Appl. Phys. Lett.* **90** (2007) 082903, *J. Appl. Phys.* **110** (2011) 052019). As introduced in the reported amorphous films of BaTiO₃ and SrTiO₃, the identical ‘random network of local bonding units (RN-LBU) theory’ is likely applicable to the amorphous CCTO films, which assumes the presence of the weak orientational ordering of TiO₆ octahedra in local scale.

Concerning defect-dipoles as observed in crystalline CCTO in literature, amorphous CCTO films can be characterized as possessing various defect-dipoles from the evidences of the presence of defects in the XPS results with oxygen partial pressure (in Fig. 1c). As seen below in the schematic, a representative example of defect dipole (marked as Defect dipole(1)), $Ti'_{Ti} - V''_o$, modifies the nature of octahedron because the oxygen vacancy and reduced Ti³⁺ substitute the lattice oxygen and Ti⁴⁺ in the octahedron. As another defect dipole (marked as Defect dipole(2)), $Cu'_{Cu} - V''_o$ (between the reduced Cu¹⁺ and oxygen vacancy) is assumed to form in the amorphous structure. These defects are commonly recognized in the crystalline CCTO (*Appl. Phys. Lett.* **90** (2007) 082903, *J. Appl. Phys.* **110** (2011) 052019), which are responsible for the interfacial polarization across the grain boundaries. Note that our amorphous structure demonstrated the existence of structural irregularities acting like short-range grain boundary from the Col-Cole plots (Fig. 3h) only for the samples at high oxygen partial pressures. Conclusively, the presence of defect dipoles incurs extra polarization at low frequencies as observed in Fig. 3g where dielectric constant is seriously raised at lower frequencies than 10^4 Hz (which is similar to the observation in the reported crystalline case).

Amorphous CCTO structure with defect dipoles (the part of this figure was included in Fig. 4d)

Concerning the mechanism of piezoelectric power generation, the local dipoles of TiO₆ distributed without long-range order are believed to be the main reason for the occurrence of electromechanical conversion, basically identical to the case of crystalline CCTO where the TiO₆ octahedra are responsible for piezoelectricity. The involvements of the defect dipoles, which are within the octahedral and connecting the octahedra, are also likely the contributors to the enhanced polarization because the bending operation extend the lengths of defect dipoles. At the same time, however, the enhanced polarization may increase dielectric constant and can act negatively to the piezoelectric power generation (because the higher dielectric constant reduces the piezoelectric voltage coefficient). Note that the figure of merit (FOM) for power generation is given by $d_{33} \times g_{33}$ where g_{33} is the piezoelectric voltage coefficient ($g_{33} = d_{33}/\epsilon_r \epsilon_0$ where ϵ_r is the relative permittivity

and ϵ_0 is the zero permittivity) However, the raised dielectric constant competes with the enhanced piezoelectricity for the higher figure of merit (FOM) as a parameter of effectiveness as power generator. The higher piezoelectric coefficient with a lower dielectric constant is demanded for the higher FOM. We calculated FOM as a function of oxygen partial pressure as seen below. As expected, the sample processed at 4.0 mTorr (which exhibited the maximum harvesting performance) showed the best FOM of $1.22 \times 10^{-12} \text{ m}^2\text{N}^{-1}$, which was substantially raised compared to $0.06 \times 10^{-12} \text{ m}^2\text{N}^{-1}$ at 1.8 mTorr, as listed below. The estimated results were included as new Supplementary Table 2.

Supplementary Table 2. Estimation of the piezoelectric voltage coefficient g_{33} and the figure of merit (FOM) for the harvesters based on amorphous CCTO thin films processed at different oxygen partial pressure. Note that we used the dielectric constant values measured at 40 kHz to match with the PFM frequency.

Oxygen partial pressure	d_{33} (pm V ⁻¹)	ϵ_r (ϵ_{33}/ϵ_0)	g_{33} (m V N ⁻¹)	FOM ($10^{-12} \text{ m}^2 \text{ N}^{-1}$)
1.8 mTorr	1.73	5	0.039	0.06
2.3 mTorr	5.16	12	0.049	0.25
3.0 mTorr	17.8	38	0.052	0.94
4.0 mTorr	27.7	71	0.044	1.22

In the regard to the discussion above, we revised/added the following description in text:

“Interestingly, there have been a few reports on dipolar polarization in cases of amorphous perovskite oxide films, e.g., amorphous BaTiO₃³⁴ and SrTiO₃^{35,36}, which experimentally demonstrated that the magnitude of Ti off-centered displacement was higher than the case of crystalline counterparts owing to the weak orientational ordering of TiO₆ octahedra in the amorphous state. Such amorphous state concerning the dipolar polarization was explained based on the random network of local bonding units (RN-LBU) theory proposing that the piezoelectricity can exist in structures that lack the spatial periodicity inherent for ionic crystals, but are composed of polar units with directional ordering in short-range scale. However, the polarity may not be greatly effective in the disordered structure, partially because the neighboring octahedra can be connected via edges and faces (not via apex-to-apex). In addition, there are rare reports on effective piezoelectric coefficients measured by PFM in the cases of amorphous perovskite oxide films, such as 16 pm V⁻¹ for 300-nm-thick KNbO₃ film³⁷ and 10 (±25%) pm V⁻¹ for 100-nm-thick SrTiO₃ film³⁶, even though the PFM results may not represent intrinsic values owing to the limitation of the PFM instrument itself^{38,39}.

(in page 3, line 14)

“Concerning the mechanism of piezoelectric power generation, the local dipoles of TiO₆ distributed with no long-range order are believed to be the main reason for the occurrence of electromechanical conversion, basically identical to the case of crystalline CCTO where the TiO₆ octahedra are responsible for piezoelectricity. The RN-LBU theory proposed in other amorphous perovskite oxide films³⁴⁻³⁶ is believed to be also applicable to this amorphous CCTO film. As observed in the XPS analysis in Fig. 1c, the defect dipoles of $Ti'_{Ti} - V_o''$ modifies the nature of TiO₆ octahedra because the oxygen vacancy and reduced Ti³⁺ substitute the lattice oxygen and Ti⁴⁺ in the octahedra. As another defect dipole of $Cu'_{Cu} - V_o''$ between the reduced Cu¹⁺ and oxygen vacancy is assumed to form in the amorphous structure. These defects are commonly recognized in the crystalline CCTO, which are responsible for the interfacial polarization across the grain boundaries⁵³⁻⁵⁵. The involvements of the defect dipoles, which are within the octahedral and connecting the octahedra,

are likely the additional contributors to the enhanced polarization because the bending operation extends the lengths of defect dipoles. At the same time, however, the enhanced polarization increases dielectric constant and can act negatively to the piezoelectric power generation (because the higher dielectric constant reduces the piezoelectric voltage coefficient). However, the raised dielectric constant competes with the enhanced piezoelectricity for the higher figure of merit (FOM) as a parameter representing the effectiveness as power generator. The higher piezoelectric coefficient with a lower dielectric constant is demanded for the higher FOM. FOM given by $d_{33} \times g_{33}$, where g_{33} is the piezoelectric voltage coefficient, was estimated after evaluating g_{33} with the relation of $g_{33} = d_{33}/\epsilon_r \epsilon_0$ where ϵ_0 is the zero permittivity^{17,56}. The estimated results are presented in Supplementary Table 2. As expected, the sample processed at 4.0 mTorr (that exhibited the maximum harvesting performance) showed the best FOM of $1.22 \times 10^{-12} \text{ m}^2 \text{ N}^{-1}$, which was substantially raised compared to $0.06 \times 10^{-12} \text{ m}^2 \text{ N}^{-1}$ at 1.8 mTorr.”

(in page 13, line 22)

2. The novelty of employing CCTO for harvesting energy is not justified. Various piezoceramics and polymer composites with piezo active fillers perform better with respect to the reported performance of the CCTO. So, the author is suggested to explain the advantages of reporting method and material as compared to other piezoelectric materials.

REPLY) As we believe, this work is very interesting because an unusual perovskite oxide holding intrinsically dipolar and space-charge polarization leads to outstanding harvesting performance for the amorphous films deposited directly on a plastic substrate using regular sputtering. Note that a very expensive transfer method (like the laser lift-off, substrate etching, or exfoliation-transfer method) was necessary to attain perovskite oxide films on a plastic substrate. Apart from the technical achievements, we also intended to pursue the origin of the unique harvesting outcomes in the amorphous films, which has never been available thus far. This result may incur subsequent studies using other stable amorphous perovskite oxide films.

In addition, we have recognized recent reports on perovskite halide materials which can process at temperature below 100 °C, which is suitable for flexible piezoelectric energy harvesters, even with the form of halide-polymer composites. Therefore, we may have to mention general disadvantages of perovskite halides, such as poor chemical stability with long exposure, limited film thickness (dissolution issue of chemical precursors with repetitive coatings), and high cost of halide chemical precursors, which make them away from serious commercialization.

(NOTE. we actually made halide film samples to compare the chemical stability with our case, as a part of the response to the third reviewer in this reply, showing the chemical degradation with atmospheric exposure. Our sample did not show such degradation after the same period of exposure time)

In fact, we re-compared our best harvesting performance with results for the piezoelectric thin-film harvesters (revised Supplementary Table 3 and Fig. 4f, as also seen below) and halide-composite harvesters (new Supplementary Table 4 and Supplementary Fig. 14) with the added parameter of power density as seen below. Surprisingly, our best values, power & power density, outperformed all the reported values in both tables.

(NOTE. We understand that the harvesting performance is very hard to be simply compared with the reported output values because the harvesting results depend on sample dimension, device structure (with electrode pattern), and measurement technique and condition.)

Revised Supplementary Table 3. Comparison of our best harvesting performance with representative results reported for piezoelectric thin-film harvesters based on various piezoelectric materials, such as perovskite oxides, ZnO, AlN, and perovskite halides.

Material (Substrate)	Deposition method	Film thickness (nm)	Poling field (kV cm ⁻¹)	Output voltage (V)	Output current (μA)	Power (μW)	Power density (μW cm ⁻³)	Mechanical input source (condition)	Ref.
Pb(Zr _{0.53} Ti _{0.47})O ₃ (Stainless steel)	Sputtering	2,800	Poled	2.6	N/A	244	1.1 × 10 ⁶	Vibration (50 Hz)	[3]
Pb(Zr,Ti)O ₃ (Si)	Spin-coating	1,000	Poled	0.16	N/A	2.15	3.2 × 10 ³	Vibration (462 Hz)	[10]
Mn-Pb(Zr _{0.52} Ti _{0.48})O ₃ (Ni foil)	Sputtering	3,000	Poled	2.44	N/A	60	1.6 × 10 ⁶	Vibration (72 Hz)	[11]
Mn-(K _{0.5} Na _{0.5})NbO ₃ (Si)	Spin-coating	1,000	120	0.52	N/A	3.6	1.8 × 10 ³	Vibration (132 Hz)	[58]
(K,Na)NbO ₃	Sputtering	2,200	Poled	0.42	N/A	1.6	4.1 × 10 ²	Vibration (393 Hz)	[59]
AlN (Si)	Sputtering	500	No poling	2.48	N/A	20.5	4.1 × 10 ³	Vibration (210 Hz)	[61]
(Bi _{0.5} Na _{0.5})TiO ₃ -BaTiO ₃ (Si)	Spin-coating	2,000	600	0.75	N/A	2.22	4.6 × 10 ⁴	Vibration (42 Hz)	[8]
PZT (Stainless steel)	Sputtering	3,000	Poled	9.4	N/A	13.4	5.3 × 10 ⁴	Pressing (finger)	[57]
ZnO (PET)	Sputtering	2,000	No poling	2.25	N/A	0.28	1.4 × 10 ³	Vibration (370 Hz)	[60]
AlN (PI)	Sputtering	900	No poling	0.7	N/A	1.4 × 10 ⁻³	4.0 × 10 ²	Bending (N/A)	[62]
CsPbBr ₃ (ITO-PEN)	Spin-coating	545	35	22.6	1.13	21.3	5.6 × 10 ⁴	Bending (strain, 0.67%)	[63]
CsSnI ₃ (ITO-PEN)	Spin-coating	354	17.4	9.5	0.45	4.23	2.4 × 10 ⁴	Bending (N/A)	[64]
MAPbI ₃ (ITO-PEN)	Spin-coating	486	11.3	23.1	1.70	182	2.7 × 10 ⁵	Bending (strain, 0.47%)	[65]
MA ₂ SnCl ₆ (ITO-PET)	Spin-coating	300	50	12	1.20	7.33	2.4 × 10 ⁵	Pressing (0.5 MPa)	[66]
Amorphous CaCu ₃ Ti ₄ O ₁₂ (PEN)	Sputtering	497	120	38.7	1.24	413	2.8 × 10 ⁶	Bending (strain, 0.77%)	This work

MA: methylammonium, PI: polyimide, PET: Polyethylene terephthalate

Revised Fig. 4f. Plot comparing our best values of voltage, power, and power density with the reported values for thin-film-based piezoelectric energy harvesters for various piezoelectric materials including ZnO, AlN, PZT, perovskite halides. Detailed information on the values cited in the plot is listed in Supplementary Table 3.

Supplementary Table 4. Comparison of our best harvesting performance with representative results reported for piezoelectric filler-polymer matrix composite harvesters.

Filler (content)	Polymer (substrate)	Thickness (μm)	Poling field (kV cm^{-1})	Output voltage (V)	Output current (μA)	Power (μW)	Power density ($\mu\text{W cm}^{-2}$)	Power density ($\mu\text{W cm}^{-3}$)	Mechanical Input sources (condition)	Ref.
PZN-PZT (30 vol.%)	UV polymer (ITO-PET)	15	75	2.96	0.36	0.81	0.17	1.1×10^2	Bending (strain, 0.85%)	[67]
BaTiO ₃ (10 wt.%)	PVDF (ITO-PET)	40	500	102*	10*	280	70	1.8×10^4	Pressing (500 kPa)	[68]
BaTiO ₃ (30 wt.%)	PDMS (Cu foil)	180	300	3.05	2.5	1	0.25	1.7×10^2	Pressing (-)	[69]
NaNbO ₃ (1 vol.%)	PDMS (PS)	100	80	3.2	0.07	27	6	6.0×10^2	Bending (strain, 0.23%)	[70]
BiFeO ₃ (40 wt.%)	PDMS (PET)	100	200	3	0.3	0.25	0.12	1.2×10^1	Pressing (finger)	[71]
CsPbBr ₂ I (10 wt.%)	PDMS (Polyester)	10	700	65	12	375	32.2	3.2×10^4	Pressing (1.3 N)	[72]
FAPbBr ₃ (35 wt.%)	PDMS (ITO-PET)	150	50	8.5	3.4	36	12	8.0×10^2	Pressing (0.5 MPa)	[73]
FAPbBr ₃ (12 wt.%)	PVDF (PET)	120	50	30	29.7	131	27.4	2.3×10^3	Pressing (0.5 MPa)	[74]
FASnBr ₃ (20 wt.%)	PDMS (ITO-PET)	80	750	94.6*	19.1*	118	18.9	2.4×10^3	Pressing (4.2 N)	[75]
MASnI ₃ (N/A)	PVDF (PI)	5	60	12	4	21.6	21.6	4.3×10^4	Pressing (0.5 MPa)	[76]
MASnBr ₃ (15 wt.%)	PDMS (PI)	172	55	18.8	13.76	74.5	74.5	4.3×10^3	Pressing (0.5 MPa)	[77]
(ATHP) ₂ PbBr ₂ Cl ₂ (30 wt.%)	PDMS (ITO-PET)	300	200	90*	6.5*	N/A	1.7	5.7×10^1	Pressing (4.2 N)	[78]
FAPbBr ₂ I (20 wt.%)	PVDF (Polyester)	30	40	85*	30*	144	10	3.3×10^3	Pressing (1.3 N)	[79]
TMCM ₂ SnCl ₆ (18 wt.%)	PDMS (Cu foil)	300	150	81	2	N/A	N/A	N/A	Pressing (4.9 N)	[80]
MAPbI ₃ (25 vol.%)	PVDF (ITO-PET)	98	80	45.6	4.7	N/A	N/A	N/A	Pressing (50 N)	[81]
Amorphous CaCu ₃ Ti ₄ O ₁₂	- (PEN)	(0.497)	120	38.7	1.24	413	138	2.8×10^6	Bending (strain, 0.77%)	This work

(PZN-PZT: $\text{Pb}(\text{Zn}_{1/3}\text{Nb}_{2/3})\text{O}_3\text{-Pb}(\text{Zr}_{0.5}\text{Ti}_{0.5})\text{O}_3$, FA: formamidinium, MA: methylammonium, ATHP: 4-aminotetrahydropyran, TMCM: trimethylchloromethylammonium, PS: polyester, PET: polyethylene terephthalate, PI: polyimide, PVDF: polyvinylidene fluoride)

*based on peak-to-peak values: half of each value needs to be considered for comparison with the other values in this Table.

Supplementary Fig. 14. Comparison of our best power and power density values with the reported values for halide-polymer composite harvesters.

3. How to confirm the film stoichiometry is exactly equal to CaCu₃Ti₄O₁₂? Why not the formation of another compound? Non-centrosymmetric nature is the basic requirement for producing the piezoelectric effect in any compound. In the present case, how amorphous CCTO is producing piezoelectricity, and how does it differ from the crystalline CCTO film?

REPLY) We did extra experiments to chase changes in stoichiometry of the amorphous films with the variations in oxygen partial pressure as seen below in the XPS spectra with a table where the atomic ratios are listed. Nearly perfect stoichiometric ratios of 1.00:3.01:3.99:12.00 for Ca:Cu:Ti:O were confirmed at the highest oxygen partial pressure of 4.0 mTorr, indicating the importance of sufficient supply of oxygen during the deposition. The stoichiometric ratios at 4.0 mTorr is closely associated with the formation of the random network of local octahedral units but not in the long-range order, which was described above in this response.

Fig. 1c. XPS spectra of the CCTO thin films deposited at pO₂ of 1.8, 3.0, and 4.0 mTorr for the binding energy regions corresponding to the Cu 2p_{3/2}, Ti 2p_{3/2}, and O 1s states.

New Supplementary Table 1. Atomic percentage and relative ratios in amorphous CCTO thin films, which were obtained from the XPS spectra in Fig. 1c.

pO ₂		Ca	Cu	Ti	O
1.8 mTorr	at.%	5.03	14.92	20.38	59.67
	Ratio	1.00	2.97	4.05	11.86
3.0 mTorr	at.%	5.02	14.88	20.18	59.92
	Ratio	1.00	2.98	4.04	11.98
4.0 mTorr	at.%	5.00	15.05	19.97	59.98
	Ratio	1.00	3.01	3.99	12.00

We added the stoichiometry information based on the new Supplementary Table 1, with the following description in text:

“The stoichiometry of the amorphous films was estimated based on the XPS analysis, as presented in Supplementary Table 1 where the changes in atomic ratios with pO₂ are listed. The higher pO₂ changed the atomic ratios toward the stoichiometric ratios of crystalline CCTO. As a result, nearly perfect stoichiometric ratios of 1.00:3.01:3.99:12.00 for Ca:Cu:Ti:O were achieved for the film processed at the highest pO₂ of 4.0 mTorr, indicating that more supply of oxygen is critical in reaching the stoichiometric ratios.”

(in page 6, line 20)

Piezoelectricity of the amorphous state is believed to originate mainly from the existence of randomly distributed TiO_6 dipoles in local regions. The following structural schematic may be appropriate in illustrating the local network of randomly distributed octahedra in the amorphous state.

As described previously in this response, the piezoelectricity is likely to be associated with the observed ferroelectricity even in the amorphous state with short-range order. Identically to the crystalline state, Ti off-centering in TiO_6 octahedra must be responsible for dipolar polarization, while we expect that the distribution of the octahedra is random and the octahedra are not connected only via apex-to-apex in the amorphous state. Because of the randomness of the octahedra networks, the overall magnitude of dipolar polarization may not be the level of crystalline case. We also have to mention that the octahedra themselves are modified with substituted defects, like oxygen vacancies and reduced Ti, and thus influence the piezoelectric response. As the form of defect dipoles, the defects may also influence the polarization even in the amorphous state.

The following schematic is a reported schematic for the amorphous perovskite oxides, demonstrating the local octahedra in random connections (not only apex-to-apex) based on the 'random network of local bonding units (RN-LBU)' theory.

Schematics of crystalline and amorphous structures of perovskite oxide reported in *Sci. Adv.* **8** (2022) eabo5977 & *Adv. Mater.* **22** (2010) 2485. Note that the octahedra can be connected via edges or faces (right)

4. The author has followed the direct growth of CCTO over ITO-coated PEN film. What is the significance of the PDMS layer over the active piezo film? The device fabrication seems defective, and the output may be influenced by PDMS triboelectrification. So, the device needs to be refabricated with the proper precaution to measure the inherent performance under the external stimulus.

REPLY) We agree that the potential gap between the PDMS and CCTO layers may create the triboelectric effect. Since we also initially doubted the potential influence by triboelectricity, we made efforts to get reliable performance through multiple-sample measurements. The PDMS layer was used to provide more stable harvesting performance with consistent outcomes over a long period of time. Note that we carefully applied two-step processing for the intimate adhesion of the PDMS layer: the pre-curing of the spin-coated PDMS layer at 120 °C for 5 min and then annealing with the contacted CCTO layer at 120 °C for 30 min. We should mention that similar energy harvesting outcomes were produced even without the PDMS layer, but the consistency of peak-to-peak values was greatly improved with the PDMS layer.

The following harvesting performance compares the harvesting outcomes with and without using the PDMS layer. Although the output voltage and current values are similar between the two cases, more stable harvesting performance is evident with the PDMS layer. The result with the PDMS layer was added as Supplementary Fig. 2.

New Supplementary Fig. 2. Comparison of output performance, **a** voltage and **b** current, of the harvesters with and without the intermediate PDMS layer.

We also confirmed that only the PDMS layer (without the active CCTO layer) did not produce significant energy harvesting performance after the double-curing process as seen below, indicating that the energy harvesting outcomes were generated dominantly by the CCTO layer. The result was included as Supplementary Fig. 3.

Supplementary Fig. 3. **a** Output voltage and **b** output current measured with only the PDMS layer (i.e., without the CCTO film), indicating the trivial contributions of the PDMS layer.

Further, we extended the bending cycle up to 11,000 cycles (beyond the previous 3,600 cycles) to ascertain the stable harvesting performance as seen below. We evaluated the extended cycling test to confirm the physical stability. The noticeable stability over the extended cycles indicates the well-maintained attachment between the PDMS and CCTO layers. Otherwise, the outcome may be deteriorated with the damaged mechanical integrity (and thus poor electrical connectivity) in the device structure or may be interrupted presumably owing to triboelectricity if the gap between the PDMS and CCTO layers is created during the extended cycling test.

Revised Fig. 2f. Stability evaluation of the output voltage over 11,000 cycles for the CCTO thin film deposited at 4.0 mTorr, measured at the bending strain of 0.77 % and the bending frequency of 2.73 Hz.

In this regard, the following paragraph was added:

“It should be mentioned that the intermediate PDMS layer was used to generate more stable energy harvesting performance as seen in Supplementary Fig. 2 where more consistent peak-to-peak output values were created with the incorporation of PDMS layer compared to the results produced by the CCTO layer without PDMS. The harvesting performance of the PDMS layer alone was trivial, as demonstrated in Supplementary Fig. 3, as it generated an output voltage and current of only ~ 0.05 V and ~ 17 nA, respectively.”

(in page 7, line 19)

5. PFM characterization is adopted to demonstrate the piezoelectric properties of CCTO film, but it is a localized concept and may not be considered for direct evidence. It should be considered for the supporting evidence along with the standard piezoelectric measurements. Moreover, the PFM results are easily influenced by electrical interference. Therefore, authors should provide piezoelectric coefficient values of CCTO film by DBLI.

REPLY) As generally known, we agree that the PFM measurement characterizing a local area may not be an appropriate tool for providing true piezoelectric properties from the quantification errors due to ill-defined field, tip-film contact mechanics, and/or sample flexure, which can give scattered values even for the identical material. As another tool to evaluate piezoelectric coefficient of thin films, optical interferometers, such as single-beam laser interferometry (SBLI) and double-beam laser interferometry (DBLI), have been utilized to chase reliable effective piezoelectric coefficients. Unfortunately, however, it is known that the DBLI has still issues on the accuracy of the measured value, including global response, substrate effect (depending on elastic nature), reduced sensitivity (due to the increased beam length), even though the contribution of substrate bending can be

erased relative to the measurement by SBLI. As an example, effective piezoelectric coefficients in the d_{33} mode were reported to be scattered from ~ 90 to ~ 185 pm V^{-1} only depending on the electrode size from 0.5 to 2.0 mm as the result of large signal measurement by DBLI for 1.9 μm -thick PZT films (*Appl. Phys. Lett.* **103** (2013) 132904). Currently as we believe, it is very hard to make sure of accurate piezoelectric coefficients particularly in the form of thin films. It is difficult to find reports on reliable piezoelectric coefficients by either SBLI or DBLI for new thin-film materials including perovskite halides or atomic scale chalcogenides.

In fact, we did the following three extra efforts to chase the evidence of piezoelectricity in the amorphous CCTO films processed with different oxygen partial pressure, although the absolute values of the piezoelectric coefficients are questionable. Probably, the word ‘effective’ in the term of effective piezoelectric coefficient implies non intrinsic values. As long as the values are compared in the same measurement condition, it may safely suggest at least the existence of piezoelectricity and the varying trends with $p\text{O}_2$. We want to believe that the following efforts are the best choices we can do within the current technical limitations in estimating proper piezoelectricity reasonably in the flexible thin films of new material.

(1) Concerning the result of PFM measurements, we agree that our values can deviate from the true values of piezoelectric coefficients mainly because of the electrostatic effects induced by the interaction between the tip and sample surface as reported in literature (*J. Appl. Phys.* **129** (2021) 185104, *Nat. Commun.* **10** (2019) 11661, and *Curr. Appl. Phys.* **17** (2017) 661). To minimize the potential errors, as the same time, the measurement conditions were carefully adjusted, which include the use of Pt/Ir-coated tip having a high spring constant of 42 N m^{-1} (to minimize the electrostatic force), applying a low frequency of 40 kHz (away from the resonance frequency of 330 kHz for the tip), and measuring the piezoresponse in the pulsed DC mode (on-field state) (to minimize the electrostatic effect).

As an extra effort, at the same time, we provided mapping images of the estimated $d_{33,eff}$ by PFM for 400 data points over an extended area of $\sim 10 \times 10 \mu\text{m}^2$ of each film as seen below. From the contrast difference between the samples, it is clear that the stronger piezoresponse is evident for the higher $p\text{O}_2$ samples. We calculated the average values of $d_{33,eff}$ from the 400 values of each film, as being 1.73 ± 0.3 , 5.16 ± 1.0 , 17.8 ± 2.6 , and 27.6 ± 4.4 pm V^{-1} for the 1.8, 2.3, 3.0 and 4.0 mTorr samples, respectively. The values were plotted in Fig. 3d.

New Fig. 3c Mapping images of the calculated $d_{33,eff}$ from the peak amplitude at +10 V, which were constructed using 400 data points over the area of $\sim 10 \times 10 \mu\text{m}^2$ of each film.

We have clearly mentioned that the obtained $d_{33,eff}$ values do not represent true values in the revised manuscript, not to give wrong impression on the $d_{33,eff}$ values. We also provided more detailed description on the PFM measurements in the Experimental section, with the following revised description in text:

“For the intuitive comparisons in the changed PR amplitudes with pO_2 over the larger area, the mapping images of effective piezoelectric coefficient $d_{33,eff}$, which were estimated from the peak amplitude at +10 V, were constructed using 400 data points for the designated area of $\sim 10 \times 10 \mu m^2$ as seen in Fig. 3c. The mapping images demonstrate quite consistency over the extended areas, ensuring the positive effect of high pO_2 on the piezoresponse. The average $d_{33,eff}$ value increased gradually with the applied oxygen pressure, specifically from $1.73 \pm 0.3 \text{ pm V}^{-1}$ for the 1.8 mTorr sample to $27.6 \pm 4.4 \text{ pm V}^{-1}$ for the 4.0 mTorr one as plotted in Fig. 3d. Note that the average $d_{33,eff}$ may still deviate from intrinsic values due to the limitations of the PFM measurement technique although the comparison of the mapping images clearly supports the pO_2 -dependent piezoresponse.”

(in page 10, line 6)

“The piezoelectricity of the CCTO films was assessed by PFM (Multimode 8, Bruker, Germany) using a conductive Pt/Ir-coated Si cantilever tip (PPP-NCHPt, Nanosensors, USA) having a spring constant of 42 N m^{-1} . The PR amplitude and phase images were obtained by scanning the film surface at 40 kHz under an AC bias of 4 V in lock-in mode. The effect of DC bias on the amplitudes was examined in the range of -10 V and $+10 \text{ V}$. The effective piezoelectric coefficient $d_{33,eff}$ were calculated from the peak amplitudes at +10 V in the amplitude-DC bias curves. Specially, the mapping images of $d_{33,eff}$ were constructed from 400 data points over an extended area of $10 \times 10 \mu m^2$ ”

(in the Methods section, page 19, line 13)

(2) Concerning characterization of piezoelectricity by optical interferometer, either the SBLI or DBLI technique may not be suitable for our flexible CCTO films on a plastic substrate because the polymer substrate is not rigid. However, we tried to measure the effective piezoelectric coefficients by depositing CCTO thin films on a Si substrate by SBLI (owned by our lab). Unfortunately, the DBLI is not available anywhere in my country (not purchased yet as I know). The measurement results for two samples deposited at 3.0 and 4.0 mTorr are seen below (as also included as Fig 3f), confirming that piezoelectricity is at least present in the amorphous state. Calculated effective d_{33} values were ~ 18.1 and $\sim 28.5 \text{ pm V}^{-1}$ for the 3.0 and 4.0 mTorr samples, respectively.

New Fig. 3f Electric displacement versus electric field for the 3.0 and 4.0 mTorr samples, measured by laser interferometry.

We understand that the values do not represent the true values mainly from the substrate bending effect. Accordingly, we mentioned clearly the limitation of the values when the values are introduced in text as following:

“To further confirm the existence of piezoelectricity in amorphous CCTO, we made extra efforts of characterizing piezoelectric coefficients using two other measurement techniques. Fig. 3f shows the electric displacements with applied electric field for the 3.0 and 4.0 mTorr films on Si, which were recorded using single-beam laser interferometry. The larger non-linear response was found in the case of 4.0 mTorr, resulting in an estimated effective d_{33} of $\sim 28.5 \text{ pm V}^{-1}$ which is higher than $\sim 18.1 \text{ pm V}^{-1}$ for the 3.0 mTorr sample. Even though there is uncertainty in the values due to uncompensated bending effect of the substrate (as the limitation of the measurement technique)^{39,51}, at least it ensures the presence of piezoelectricity in the amorphous state. As a reference, 1.9 μm -thick crystalline PZT films reported effective d_{33} of 90 to 185 pm V^{-1} as the result of measurement using a double-beam laser interferometer³⁸.”

(in page 11, line 22)

“Electric field-dependent displacement was measured using an aixACCT TF analyzer 2000E (aixAcct Systems GmbH, Germany) equipped with a single-beam laser interferometer in the condition of $\pm 3 \text{ V}$ triangular excitation at 10 Hz for the films deposited on Si. From the displacement data, effective piezoelectric coefficient was estimated by the linear regression from point N/4 to below N/2 at the positive side, where N is the total number of measurement points.”

(in the Methods section, page 19, line 21)

(3) Measurement of effective transverse piezoelectric coefficient, $-e_{31,eff}$

Further, we made another effort to ascertain the presence of piezoelectricity by measuring effective transverse piezoelectric coefficient $-e_{31,eff}$ of the amorphous CCTO films deposited at 4.0 mTorr on Si using a 4-point bender measurement unit (aix4PB, aixACCT, Germany). We obtained a $-e_{31,eff}$ value of $\sim 1.1 \text{ C m}^{-2}$, which indicates the presence of piezoelectricity in the amorphous CCTO films. As references, $-e_{31,eff}$ values, $\sim 3.2 \text{ C m}^{-2}$ for crystalline $\text{Bi}(\text{Na}_{0.5}\text{Ti}_{0.5})\text{O}_3\text{-BaTiO}_3$ thin films (ACS Appl. Mater. Interfaces 11 (2019) 13244) and $\sim 13\text{-}16 \text{ C m}^{-2}$ for crystalline PZT films (ACS Appl. Mater. Interfaces 9 (2017) 18904 / J. Appl. Phys. 123 (2018) 014103) were reported. This method does not count the substrate bending effect. We added the following brief description on this result:

“Further, the effective transverse piezoelectric coefficient $-e_{31,eff}$ was obtained using a commercial 4-point bender unit for the 4.0 mTorr film on Si to verify the anticipated piezoelectric response without the bending effect of substrate. A $-e_{31,eff}$ value of $\sim 1.1 \text{ C m}^{-2}$ was attained for the 4.0 mTorr sample. As references, $-e_{31,eff}$ values of $\sim 3.2 \text{ C m}^{-2}$ for crystalline $\text{Bi}(\text{Na}_{0.5}\text{Ti}_{0.5})\text{O}_3\text{-BaTiO}_3$ thin films⁸ and $\sim 13\text{-}16 \text{ C m}^{-2}$ for crystalline PZT films^{2,39} were reported.”

(in page 12, line 6)

“In addition, the effective transverse piezoelectric coefficient $-e_{31,f}$ was measured at 1 Hz using a four-point bending measurement unit (aix 4PB, aixAcct Systems GmbH, Germany) in combination with the TF analyzer for the films on Si.”

(in the Methods section, page 20, line 1)

6. Author reported that the $d_{33,eff}$ values of CCTO film increased from 2.1 pm/V to 35.3 pm/V. What is the reason behind such drastic improvement of $d_{33,eff}$ on varying the partial oxygen pressure?

REPLY) As described earlier in this response, the nearly stoichiometric atomic ratios achieved at the higher oxygen partial pressure is primarily responsible for the enhancement of piezoelectricity, which are close to the crystalline structure in short range. Piezoelectricity in this amorphous state, characterized by PFM, is believed to come from the dipolar polarization of TiO_6 dipoles distributed randomly in a network of the dipole units. The contribution by the local TiO_6 octahedra becomes more effective with high $p\text{O}_2$ where oxygen vacancy and cation defects are minimized, which are acting favorably for extra dipolar polarization. With the deviations from the stoichiometry at low $p\text{O}_2$, the network of dipole units is likely less effective in creating the converse electromechanical behavior.

Regarding the increase in the piezoelectricity with the $p\text{O}_2$, we added the following description:

“Note that nearly stoichiometric atomic ratios of CCTO were achieved at the highest oxygen partial pressure of 4.0 mTorr, which is close to the crystalline structure in short range. Piezoelectricity in this amorphous state is believed to come mainly from the dipolar polarization of TiO_6 octahedra distributed randomly in a network of the dipole units. The potential domain reversal of the 4.0 mTorr sample was examined to ascertain ferroelectricity, as shown in Fig. 3e, where large contrast changes in the PR phase images of the designated boxes were observed by consecutively switching between biases of +10 and -10 V. The apparent ferroelectricity at the high $p\text{O}_2$ is believed to be associated with the dipole moments driven by local TiO_6 octahedra even in the amorphous state, which become effective with the reduced oxygen vacancies and the changed chemical states of cations with the higher $p\text{O}_2$ ^{47,48}. The existence of oxygen vacancies negatively affects polarization by acting as clamping centers against the movement of domain walls^{32,49,50}. The increased concentration of Ti^{4+} relative to Ti^{3+} with the higher $p\text{O}_2$ may produce greater polarization because the smaller ionic radius of Ti^{4+} (0.605 Å) than that of Ti^{3+} (0.67 Å) may induce a larger dipole moment³⁰.”

(in page 11. line 8)

7. The PENG device portrays some noisy peaks in the output voltage and current. The peak-to-peak output gradually increases but stable performance enhancement is not observed in the films grown at various oxygen partial pressure.

REPLY) As pointed out, we re-measured all the corresponding harvesting performance with newly fabricated harvesters with extra cautions to get more stable outcome values (including the harvesting performance with changing oxygen partial pressure). As a result, we obtained more reliable data that are basically identical to the values measured previously, as seen below. All the figures were used to replace the old versions in the revised manuscript.

Revised Fig. 2b,c Harvesting performance with changing oxygen partial pressure

Revised Supplementary Fig. 4 Harvesting performance with changing the film thickness of CCTO

Revised Supplementary Fig. 5a,b Harvesting performance with changing bending strain

Revised Supplementary Fig. 5c,d Harvesting performance with changing bending frequency

8. The stability is carried out at a single frequency. The author needs to discuss the impact of various frequencies over the piezo film or needs to provide the frequency optimization for the enhanced output of the PENG device.

REPLY) We provided the frequency-dependent output voltage and current in the bending frequency range of 0.86 to 3.50 Hz for the optimal harvesters of 4.0 mTorr CCTO film as seen below (as also included as Supplementary Fig. 5c,d). The frequency of 3.1 Hz was an optimal one for the harvesting performance.

Revised Supplementary Fig. 5c,d Harvesting performance with changing bending frequency

In addition, we evaluated the cycling stability with the extension of bending cycles up to 11,000 cycles (beyond the previous 3,600 cycles) at two different bending frequencies of 2.73 and 2.90 Hz for the optimal harvesters, as seen below. The stable performance was observed at the frequencies, with expected lower voltages at the lower frequency of 2.73 Hz.

Revised Supplementary Fig. 7. Stability evaluation of the output voltage over 11,000 cycles for the CCTO thin film deposited at 4.0 mTorr, measured at the bending strain of 0.77 % for two bending frequencies of 2.73 and 2.90 Hz.

9. The author has followed electrode poling for 60-80 kV/cm to optimize the performance of the PENG device. The electrode poling of 80 kV/cm does not provide any significant performance improvement even after the application of very high voltage.

REPLY) To confirm the effect of electric poling on the performance of the harvesters, we measured the harvesting performance again after poling up to an extended field of 120 kV cm^{-1} , as seen below. The output voltage and current values were raised to 38.7 V and 1,238 nA, respectively, at the maximum field of 120 kV cm^{-1} , which correspond to the increments by $\sim 76\%$ and $\sim 37\%$ compared to the values (22.0 V and 906 nA) attained without poling. Unlikely to the crystalline CCTO, all the TiO_6 octahedra may not only be connected via apex-to-apex, but also via edges and faces in the amorphous state (as described previously here). Accordingly, the alignment of dipoles along the poling direction may not be great as much as observed in the crystalline perovskite. Nonetheless, the levels of increments in harvesting outcomes are still significant when considering the amorphous structure. We believe that the enhancements in output performance with poling support partially the octahedral dipole-driven piezoelectricity even in the amorphous state.

Revised Fig. 4 a,b Output voltage and current obtained with the ~ 497 -nm-thick CCTO harvester after applying the electric fields, which were measured under the optimal conditions of 0.77% and 3.10 Hz. **c** Plots of peak voltage and current with increasing poling field.

10. The overall physicochemical and energy harvesting performance is inadequate to confirm the origin of piezoelectricity in amorphous CCTO film.

REPLY) As we believe, we did our best to demonstrate the evidence of piezoelectricity using available analytical techniques as described in the response above. Despite of the recognized intrinsic limitations in all the direct measurement tools for piezoelectric coefficients, we still ensure the presence of piezoelectricity in amorphous CCTO films as also conjectured from the similar observations reported in amorphous SrTiO₃ and BaTiO₃ thin films. Our result may be another example supporting the “random network of local bonding units (RN-LBU) theory” where the localized octahedral dipoles are effectively working in a short range.

Initially, we thought that demonstrating a piezoelectric device with viable performance was a practical way of proving the existence of strong piezoelectricity in the amorphous state because characterizing piezoelectric coefficients of new thin films is very hard to be satisfactorily done owing to the limited measurement techniques. Surprisingly, the harvesting performance was so excellent, outperforming the reported harvesting outcomes for perovskite thin-film harvesters (Supplementary Table 3) and even for polymer harvesters (Supplementary Table 4) in power and power density, with the merit of direct deposition on a plastic substrate without the annealing process. This study does not hold true meanings only from the energy harvesting excellency but more toward the first systematic demonstration of power generation in the state of amorphous films, particularly having defect dipoles intrinsically.

Reviewer #3:

This work developed a flexible amorphous thin-film energy harvester based on perovskite $\text{CaCu}_3\text{Ti}_4\text{O}_{12}$ (CCTO) thin films on a plastic substrate. They attributed the high-performance output of this piezoelectric nanogenerator to the unusually high permittivity of amorphous CCTO film, which was caused by interfacial and dipolar polarization mechanisms depending on the defect states and chemical migration under the bending operation. Evidences of XPS, PFM, impedance, and dielectric measurements were also provided to confirm their hypothesis.

However, there are some important questions to be clarified :

1. According to the authors' claim, the resultant amorphous nature of the films resulted in an output voltage and current of ~ 35.5 V and ~ 1.1 μA , respectively, which tops the previously reported record for typical polycrystalline ferroelectric oxide thin-film cantilevers. What are the advantages of this perovskite oxide compared to halide perovskites? Compared to this CCTO thin film, halide perovskite film can be prepared by a solution-based method at room temperature (suited for flexible substrates), which is much easier and more convenient than RF magnetron sputtering in this work. On the other hand, some halide perovskites have shown better output performance than this work. For example, Jella et al. reported a MAPbI_3 -PVDF composite-based PENG with higher outputs of 45.6 V and 4.7 $\mu\text{A}/\text{cm}^2$ (*Nano Energy* 2018, 53, 46–56); Khan et al. reported a porous FAPbBr_2 -PVDF composite-based PENG with higher outputs of 85 V and 30 μA (*J. Mater. Chem. A*, 2020, 8, 13619–13629) and a ferroelectric $(\text{ATHP})_2\text{PbBr}_2\text{Cl}_2$ -PVDF composite with high outputs of 90V and 6.5 μA (*Nano Energy* 2021, 86, 106039); Huang et al. also designed a ferroelectric halide perovskite $\text{TMCM}_2\text{SnCl}_6$ with a large d_{33} of 137 pC/N and g_{33} of 0.98 V·m/N, exhibiting high output performance of 81 V and 2 μA (*ACS Energy Lett.* 2021, 6, 16–23). All these halide perovskites show both great output performance and flexibility.

REPLY) As mentioned, perovskite halide-based thin films have an advantage of processing at low temperatures below 100 °C, which is suitable for flexible harvesters on a plastic substrate. Owing to their relatively low dielectric constant (usually less than 70) and thus high piezoelectric voltage coefficient, the application as a piezoelectric energy harvester may be suitable as reported in recent literature. When considering the polymer-matrix-based composites incorporating the piezoelectric halide fillers (as particles in most cases), some outstanding output voltage and current values were reported as exemplified in the above-mentioned four articles (*Nano Energy* 2018, 53, 46 / *J. Mater. Chem. A*, 2020, 8, 13619 / *Nano Energy* 2021, 86, 106039 / *ACS Energy Lett.* 2021, 6, 16). As generally recognized, we may need to compare the harvesting performance in terms of power and power density (normalized per area or volume) because sample dimension, device structure (plus electrode pattern), and measurement technique and condition are different per each case. Unfortunately, there is no standardized guideline of measurement for the harvesting outcomes for direct comparison.

As seen below with two additional tables, we collected the recent published articles on (1) halide-based thin film harvesters and (2) halide-based polymer composite films harvesters (including the four articles mentioned above). Some of the articles did not provide sufficient information on power or power density (even for estimation). Conclusively, as compared, there was no report outperforming our power and power density (both per area and volume) even only some of the composite cases demonstrated the higher voltage and current. Note that some of maximum voltage and current in the literature were based on the peak-to-peak values. Therefore, we need to divide the values by two (nearly) for comparison, as marked in the Table. For example, the reported peak-to-peak value of 90 V for the ferroelectric $(\text{ATHP})_2\text{PbBr}_2\text{Cl}_2$ -PVDF composite (*Nano Energy* 2021, 86,

106039) needed to be corrected approximately as ~ 45 V for comparison with other values.

(1) Comparison with the halide thin-film harvesters

Piezoelectric energy harvesting performance of perovskite halide thin film-based harvesters

(The information in this table was added as a part of Supplementary Table 3)

Material (Substrate)	Film thickness (nm)	Deposition method	Poling field (kV/cm)	Output voltage (V)	Output current (nA)	Power (μ W)	Power density (μ W cm^{-2})	Power density (μ W cm^{-3})	Mechanical input sources (condition)	Ref.
CsPbBr ₃ (ITO-PET)	260	Spin-coating	24.9	16.4	604	N/A	N/A	N/A	Bending (strain, 1.67%)	[1]
CsPbBr ₃ (ITO-PEN)	545	Spin-coating	35	22.6	1,130	21.3	3.04	5.6×10^4	Bending (strain, 0.67%)	[2]
CsSnI ₃ (ITO-PEN)	354	Spin-coating	17.4	9.5	445	4.23	0.85	2.4×10^4	Bending (finger)	[3]
MAPbI ₃ (ITO-PEN)	486	Spin-coating	11.3	23.1	1,703	182	13	2.7×10^5	Bending (strain, 0.47%)	[4]
MAPbI ₃ (PET)	520	Spin-coating	12	1.47	560	N/A	N/A	N/A	Pressing (0.2 MPa)	[5]
MA ₂ SnCl ₆ (ITO-PET)	300	Spin-coating	50	12	1,200	7.33	7.33	2.4×10^5	Pressing (0.5 MPa)	[6]
Amorphous CaCu ₃ Ti ₄ O ₁₂ (PEN)	497	Sputtering	120	38.7	1,238	413	138	2.8×10^6	Bending (strain, 0.77%)	This work

(MA: methylammonium)

[1] *Energy Environ. Sci.* **13**, 2077-2086 (2020)

[2] *Adv. Energy Mater.* **12**, 2103329 (2022).

[3] *Nano Energy* **92**, 106785 (2022).

[4] *Adv. Sci.* **10**, 2204462 (2023).

[5] *Nano Energy* **52**, 11-21 (2018).

[6] *Nano Energy* **107**, 108148 (2023).

For intuitive comparison, we plotted the power and power density values for each case as seen below, suggesting that our values are better than the others.

Comparison of our best harvesting outcomes with the reported **a** power and **b** power density values of perovskite halide-thin-film harvesters

The reported results for the halide thin-film harvesters were incorporated as a part of Supplementary Table 3, where the harvesting outcomes for all other piezoelectric thin-film harvesters based on PZT, (Na,K)NbO₃, Bi(Na,Ti)O₃, AlN, and ZnO are compared.

(2) Comparison with the halide-polymer composite harvesters

Comparison of our best harvesting performance with representative results reported for perovskite halide-based polymer composite harvesters.

(The information in this table was added as a part of Supplementary Table 4)

Filler (content)	Polymer (substrate)	Thickness (μm)	Poling field (kV cm ⁻¹)	Output voltage (V)	Output current (μA)	Power (μW)	Power density (μW cm ⁻²)	Power density (μW cm ⁻³)	Mechanical Input sources (condition)	Ref.
CsPbBr ₂ I (10 wt.%)	PDMS (Polyester)	10	700	65	12	375	32.2	3.2 × 10 ⁴	Pressing (1.3 N)	[72]
FAPbBr ₃ (35 wt.%)	PDMS (ITO-PET)	150	50	8.5	3.4	36	12	8.0 × 10 ²	Pressing (0.5 MPa)	[73]
FAPbBr ₃ (12 wt.%)	PVDF (PET)	120	50	30	29.7	131	27.4	2.3 × 10 ³	Pressing (0.5 MPa)	[74]
FASnBr ₃ (20 wt.%)	PDMS (ITO-PET)	80	750	94.6* (peak to peak)	19.1* (peak to peak)	118	18.9	2.4 × 10 ³	Pressing (4.2 N)	[75]
MASnI ₃ (N/A)	PVDF (PI)	5	60	12	4	21.6	21.6	4.3 × 10 ⁴	Pressing (0.5 MPa)	[76]
MASnBr ₃ (15 wt.%)	PDMS (PI)	172	55	18.8	13.76	74.5	74.5	4.3 × 10 ³	Pressing (0.5 MPa)	[77]
(ATHP) ₂ PbBr ₂ Cl ₂ (30 wt.%)	PDMS (ITO-PET)	300	200	90* (peak to peak)	6.5* (peak to peak)	N/A	1.7	5.7 × 10 ¹	Pressing (4.2 N)	[78]
FAPbBr ₂ I (20 wt.%)	PVDF (Polyester)	30	40	85* (peak to peak)	30* (peak to peak)	144	10	3.3 × 10 ³	Pressing (1.3 N)	[79]
TMCM ₂ SnCl ₆ (18 wt.%)	PDMS (Cu foil)	300	150	81	2	N/A	N/A	N/A	Pressing (4.9 N)	[80]
MAPbI ₃ (25 vol.%)	PVDF (ITO-PET)	98	80	45.6	4.7	N/A	N/A	N/A	Pressing (50 N)	[81]
Amorphous CaCu ₃ Ti ₄ O ₁₂	- (PEN)	(0.497)	120	38.7	1.24	413	138	2.8 × 10 ⁶	Bending (strain, 0.77%)	This work

(FA: formamidinium, MA: methylammonium, PET: Polyethylene terephthalate, PI, polyimide, ATHP: 4-aminotetrahydropyran, TMCM: trimethylchloromethylammonium, PVDF: polyvinylidene fluoride)

*based on peak-to-peak values: half of each value needs to be considered for comparison with the other values in this Table.

[72] *Small* **19**, 2303366 (2023)

[73] *Adv. Funct. Mater.* **26**, 7708-7716 (2016)

[74] *Nano Energy* **37**, 126-135 (2017)

[75] *Nano Energy* **101**, 107631 (2022)

[76] *Nano Energy* **57**, 911-923 (2019)

[77] *ACS Appl. Mater. Interfaces* **12**, 16469-16480 (2020)

[78] *Nano Energy* **86**, 106039 (2021)

[79] *J. Mater. Chem. A* **8**, 13619 (2020)

[80] *ACS Energy Lett.* **6**, 16-23 (2021)

[81] *Nano Energy* **53**, 46-56 (2018)

Also for intuitive comparison, we plotted the power and power density values for each case as seen below, suggesting that our values are better than the others.

New Supplementary Fig. 14. Comparison of our best power and power density values with the reported values for halide-polymer composite harvesters

With the addition of the harvesting information on perovskite halides, we added the following description in text:

“In addition, our best harvesting performance was compared to the power-generation outcomes reported for polymer-matrix composite harvesters where piezoelectric materials such as PZT, BaTiO₃, and perovskite halides are incorporated as fillers in various polymer matrix as listed in Supplementary Table 4⁶⁷⁻⁸¹. Our best values of 413 μW and 2.8 × 10⁶ μW cm⁻³ are still better than the values achieved for piezoelectric composites even though some of the composite harvesters demonstrated outstanding output voltage and current values greater than our values (see Supplementary Fig. 14 for each comparison)^{75,78-80}. The highest power density value of 4.3 × 10⁴ μW cm⁻³ was reported for the MASnI₃- polyvinylidene fluoride (PVDF) composites⁷⁶, which is two-order lower than our best value.”

(in page 17, line 6)

Further, we may have to mention general disadvantages of perovskite halides, such as poor chemical stability with long exposure, limited film thickness (dissolution issue of chemical precursors with repetitive coatings), and high cost of halide chemical precursors, which make them away from serious commercialization. This work is very interesting because an unusual perovskite oxide holding intrinsically dipolar and space-charge polarization leads to outstanding harvesting performance in the amorphous films deposited directly on a plastic substrate using regular sputtering. Note that a very expensive transfer method (like the laser lift-off method) was necessary to attain perovskite oxide films on a plastic substrate. Apart from the technical achievements, we also intended to pursue the origin of the unique harvesting mechanism in the amorphous films. This result may incur subsequent studies using other stable amorphous perovskite oxide films.

As a reference, we conducted a simple test chasing the chemical stability of MAPbI₃ (MA: methyl ammonium) (as a representative organic-inorganic hybrid halide) and CCTO films after 15 days as seen below. Partial decomposition of MAPbI₃ films into PbI₂ was observed after 15 days' atmospheric exposure as seen in the XRD patterns, whereas the 4 mTorr CCTO film was stable after 15 days as no changes were observed in the XPS spectra of the constituent ions (we used XPS (instead of XRD) because the CCTO is amorphous).

Comparison of chemical stability between the MAPbI₃ and CCTO thin films after 15 days' atmospheric exposure: **a** XRD patterns of MAPbI₃ films showing a decomposed phase of PbI₂ after 15 days and **b** no changes in XPS spectra of Ti 2p_{3/2} and O1s peaks after 15 days.

2. The author claimed that oxygen partial pressure pO₂ was a very important parameter that affected the harvesting performance of the device. How about the stability of this device in an ambient environment? What is the durability of the performance that can be maintained at ambient condition? In addition, whether the performance will be affected if the device was put into a high-vacuum condition.

REPLY) As suggested, we did extra experiments to confirm the stability in ambient atmosphere after 15 days using the XPS results showing the changes in chemical states of ions as seen below. There were no actual changes in chemical states after 15 days for the CCTO samples processed at 1.8 and 4.0 mTorr, indicating strong chemical stability of the CCTO films.

New Supplementary Fig. 8. XPS spectra of the CCTO thin films deposited at **a** 1.8 and **b** 4.0 mTorr after 15 days' exposure in ambient atmosphere (relative to the case of the first day) for the chemical states of Cu 2p_{3/2}, Ti 2p_{3/2}, and O1s.

In addition, we also evaluated the energy harvesting performance after 15 days' atmospheric exposure of the optimal device as seen below. As expected, no significant performance degradation was observed.

New Supplementary Fig. 9. Comparison of output performance, **a** voltage and **b** current, of the 4.0 mTorr harvesters after 15 days' exposure in ambient atmosphere

For the confirmation of longer lifetime, we also extended the bending repetitive operations up to 11,000 cycles (instead of previous 3,600 cycles) at two bending frequencies of 2.73 and 2.90 Hz as also seen below. The performance was well maintained over the extended bending cycles.

Revised Supplementary Fig. 7. Stability evaluation of the output voltage over 11,000 cycles for the CCTO thin film deposited at 4.0 mTorr, measured at the bending strain of 0.77 % for two bending frequencies of 2.73 and 2.90 Hz.

Further, we placed the CCTO films in a high vacuum condition of 6×10^{-6} mbar for 48 hours and then evaluated the chemical states of the constituent ions by XPS as seen below. We did not see any noticeable changes in the chemical states after the high vacuum exposure.

XPS spectra of the CCTO thin films deposited at 4.0 mTorr after storing in a high vacuum condition of 6×10^{-6} mbar for 48 hours, relative to the case of as-deposited, for the chemical states of Cu $2p_{3/2}$, Ti $2p_{3/2}$, and O1s.

In this regard, we added the following description on the CCTO stability:

“We also confirmed the stability of the harvesting performance by continuing bending up to 11,000 cycles, as shown in Fig. 2f where consistently stable output values were attained over the extended cycles at 2.73 Hz for the ~497-nm-thick film. Supplementary Fig. 7 shows another case of the harvesting stability at a higher bending frequency of 2.90 Hz for the identical sample, confirming that the physical integrity is well preserved during repetitive bending operations. Chemical stability of the amorphous CCTO films was evaluated in terms of changes in chemical states of ions after exposing the films in ambient atmosphere for 15 days as compared in the XPS spectra of Supplementary Fig. 8 before and after the exposure. As expected, no noticeable chemical changes were observed with no degradation in the harvesting performance after the exposure period as presented in Supplementary Fig. 9.”

(in page 9, line 6)

3. Would the ambient humidity affect the stability and performance of this device? Evaluation as a function of humidity would be required.

REPLY) As requested, we conducted extra experiments by placing the optimal harvesters in a humidity chamber, with the changes of the humidity level from 30 to 80 %, and then investigating energy harvesting performance in the optimal measurement condition as seen below. We found that the level of humidity did not influence the harvesting outcome.

New Supplementary Fig. 10. Output voltage generated by an optimal CCTO harvester performance with the changes in humidity level as measured at bending strain of 0.77% and bending frequency of 2.90 Hz.

In this regard, we added the following sentence in text:

“As another effort, the harvesting performance was monitored with the changes in humidity level from 30 to 80 % as seen in Supplementary Fig. 10, suggesting that the optimal harvester device is quite stable at the humid atmosphere.”

(in page 9, line 15)

4. As for Figure 3d, the authors claimed that the dipolar polarization contributed up to ~75 to the permittivity value for 4.0 mTorr sample, which was determined by following the horizontal line to the y-axis in Fig.3d. From this figure, we can also notice that a sharp increase and decrease of dielectric constant existing at the ~10⁵ frequency. The authors should provide more details to

explain why the dielectric constant change in this manner. How does the dipolar polarization affect the shape of this dielectric curve? Are there any different interactions existing in this frequency?

REPLY) As mentioned, the distinguishable change in dielectric constant at 10^5 Hz (as marked with a red box in the curve below in Fig. 3g) is noticeable. The dielectric resonance behavior at the frequency indicates the presence of dipolar polarization with permanent dipoles as the evidence of ferroelectricity, which originates from the off-centered Ti ions in TiO_6 octahedra as observed typically in ferroelectric perovskite oxides. Note that the dielectric resonance peak at $\sim 10^5$ Hz due to the dipolar polarization becomes more prominent with the highest oxygen pressure of 4.0 mTorr (where the stoichiometric ratios among the CCTO constituents are preserved) compared to the resonance behavior at low oxygen partial pressures.

Revised Fig. 3g $p\text{O}_2$ -dependent changes in dielectric dispersion of amorphous CCTO thin films in the frequency range of $\sim 10^2$ to 10^6 Hz, suggesting the frequency-dependent contributions by different polarization mechanisms (divided into the sections with the horizontal lines) with the red box highlighting the dielectric resonance due to the dipolar polarization at $\sim 10^5$ Hz.

With the critical observation of the dielectric resonance at $\sim 10^5$ Hz as the evidence of existence of ferroelectricity, we described the significance of the dielectric behavior more clearly as following:

“The frequency dependence of the dielectric dispersion is highlighted by two distinguishable characteristics, i.e., the abrupt increases in ϵ_r toward the lower frequency in the frequency range below $\sim 10^4$ Hz and the dipolar resonance peaks at $\sim 10^5$ Hz (as highlighted with a red box in Fig. 3g). These characteristics were driven by two polarization mechanisms having different dipole moments at specific frequencies: interfacial (or space-charge) polarization at frequencies below 10^4 Hz and dipolar (or ferroelectric) polarization in the frequency range of $\sim 10^4$ to $\sim 10^5$ Hz, as guided with the horizontal lines in Fig. 3g. Note that crystalline CCTO is known to possess intrinsic interfacial polarization mainly across grain boundaries, which is driven by defects including oxygen vacancies and resultant defect dipoles^{31,47}. This large contribution by the interfacial polarization is meaningful when considering no grain boundaries present in the amorphous state. The dielectric resonance at $\sim 10^5$ Hz became clearer in the case of 4.0 mTorr, which must be related to the stronger ferroelectric nature of the films prepared with a higher $p\text{O}_2$, although the contribution of interfacial polarization to the relative permittivity is larger than that of dipolar polarization when the frequency becomes lower.”

(in page 12, line 15)

REVIEWER COMMENTS

Reviewer #1 (Remarks to the Author):

The authors have meticulously addressed the major comments that I pointed out by thoroughly reconstructing the figures and tables through additional experiments, illustrations, and references in the revised manuscript. Therefore, I believe this manuscript is qualified enough to be published in Nature Communications.

Reviewer #2 (Remarks to the Author):

The author has revised the manuscript and conducted additional experiments to support the findings. The overall scientific discussion on piezoelectricity generation has improved, though a few issues need to be clarified further. The author can clarify the following comments before making any final decision.

Comments:

1. The author has discussed the dipolar polarization and defect dipoles along the piezoelectric power generation. However, the author could compare the parameters with other fabrication techniques over the proposed process in this work.
2. The author has proposed the piezoelectricity generated from the randomly oriented TiO₆ dipoles. How about the reliabilities of the proposed structure to induce piezoelectricity over each fabricated film? If the dipoles are randomly distributed, the output may not be constant for mass production or any reliable applications, the author can discuss the dipole distribution.
3. The author has provided the PENG device output with and without the PDMS layer. In the case of PDMS, the output peaks are more stable. The author needs to clarify what encapsulation was used for the device without a PDMS layer, as without any antistatic protection, the external impact layer influences the output.
4. The author needs to justify the drastic increment of the piezoelectric, whether the randomly oriented dipoles are aligned and contributing to enhancement or decrement of the piezoelectric performance.
5. The author has provided the influence of the oxygen pressure, thickness, strain, bending strain, and bending frequency. The linear increment of output is observed in all the parameters. However, is there any standard common correlation between these factors contributing to performance enhancements?
6. The material and concept of piezoelectricity are described with various experimental tools. The author must provide insight into the fabrication of high-performance PENG and some methods to reduce the higher poling voltage for a sustainable and reliable energy harvesting technique.

Reviewer #3 (Remarks to the Author):

The authors have addressed most of the concerns, and the quality of the manuscript has been improved with the additional experiments and analyses.

2nd Response to the reviewers' comments

Nature Communications manuscript: NCOMMS-23-11198A

TITLE: High-Performance Piezoelectric Energy Harvesting in Amorphous Perovskite Thin Films Deposited Directly on a Plastic Substrate

REVIEWER COMMENTS

Reviewer #1 (Remarks to the Author):

The authors have meticulously addressed the major comments that I pointed out by thoroughly reconstructing the figures and tables through additional experiments, illustrations, and references in the revised manuscript. Therefore, I believe this manuscript is qualified enough to be published in Nature Communications.

REPLY) We appreciate your positive comment.

Reviewer #2 (Remarks to the Author):

The author has revised the manuscript and conducted additional experiments to support the findings. The overall scientific discussion on piezoelectricity generation has improved, though a few issues need to be clarified further. The author can clarify the following comments before making any final decision.

Comments:

1. The author has discussed the dipolar polarization and defect dipoles along the piezoelectric power generation. However, the author could compare the parameters with other fabrication techniques over the proposed process in this work.

REPLY) We compared the parameters, dipolar polarization and defect dipoles, with the results reported for CCTO films prepared by other fabrication techniques over our observations. Please note that the case of amorphous CCTO thin films was not reported by other research groups. In fact, the dipolar polarization (ferroelectricity) and defect dipoles have been identically recognized in reported crystalline CCTO thin films. We described the parameters in the text, in comparison with the results reported for polycrystalline CCTO films processed by other techniques including solution deposition and pulsed laser deposition (and also for the bulk CCTO).

In this regard, the following sections correspond to the description on the parameters (we added five more references for the extended discussion):

“Piezoelectricity in this amorphous state is believed to come mainly from the dipolar polarization of TiO_6 octahedra distributed randomly in a network of the dipole units. The evidence of dipolar polarization (or ferroelectricity) has been reported in polycrystalline CCTO thin films processed by various deposition techniques^{30-32,47,48,54}.”

(in page 11, line 11)

“Note that polycrystalline CCTO is known to possess intrinsic interfacial polarization mainly across grain boundaries, which is driven by defects including oxygen vacancies and resultant defect dipoles, as reported elsewhere^{30,31,41,47}.”

(in page 12, line 25)

“As observed in the XPS analysis in Fig. 1c, the defect dipoles of $Ti'_{Ti} - V''_O$ modifies the nature of TiO_6 octahedra because the oxygen vacancy and reduced Ti^{3+} substitute the lattice oxygen and Ti^{4+} in the octahedra. As another defect dipole of $Cu'_{Cu} - V''_O$ between the reduced Cu^{1+} and oxygen vacancy is assumed to form in the amorphous structure. These defects are commonly recognized in the crystalline CCTO, which are responsible for the interfacial polarization across the grain boundaries⁵⁵⁻⁵⁷. Identical defect dipoles were reported as being responsible for the enhanced permittivity in polycrystalline CCTO thin films processed by other deposition techniques including solution deposition⁵⁸ and pulsed laser deposition^{59,60},”

(in page 14, line 4)

2. The author has proposed the piezoelectricity generated from the randomly oriented TiO_6 dipoles. How about the reliabilities of the proposed structure to induce piezoelectricity over each fabricated film? If the dipoles are randomly distributed, the output may not be constant for mass production or any reliable applications, the author can discuss the dipole distribution.

REPLY) The random distribution of TiO_6 octahedra is a part of the nature of amorphous structure. For example, the TiO_6 octahedra are connected via apex-to-apex in the case of crystalline films, while they can be connected via edges and faces (as described in our manuscript). Because only the short-range order exists in the amorphous structure, piezoelectricity is assumed to be much less in the case of amorphous films (compared to the crystalline counterpart).

As we believe, however, we guess that the randomly distributed dipoles do not necessarily induce the poor reliability of the harvesting performance since the overall structure is consistent even in the amorphous state (For example, glass has consistent properties even in the amorphous state). Please note that the sputtering technique itself is an acceptable technique for mass production of films. We verified the reliable harvesting performance with extended bending cycles in various conditions: after 15 days' ambient exposure, different bending frequencies, different humidity levels, and bending cycles after poling. The reliability of the amorphous films was confirmed in all the cases (as seen below)

We provided the following results concerning the reliability of the harvesting performance:

(1) Reliability of the harvesting performance after 15 days

We did extra experiments to confirm the stability in ambient atmosphere after 15 days by evaluating the energy harvesting performance after 15 days' atmospheric exposure of the optimal device as seen below. As expected, no significant performance degradation was observed.

Supplementary Fig. 9. Comparison of output performance, **a** voltage and **b** current, of the 4.0 mTorr harvesters after 15 days' exposure in ambient atmosphere

(2) Reliability of the harvesting performance at different bending frequencies

For the confirmation of longer lifetime, we also extended the bending repetitive operations up to 11,000 cycles at two bending frequencies of 2.73 and 2.90 Hz as also seen below. The performance was well maintained over the extended bending cycles.

Supplementary Fig. 7. Stability evaluation of the output voltage over 11,000 cycles for the CCTO thin film deposited at 4.0 mTorr, measured at the bending strain of 0.77 % for two bending frequencies of 2.73 and 2.90 Hz.

(3) Reliability of the harvesting performance at different humidity levels

We conducted extra experiments by placing the optimal harvesters in a humidity chamber, with the changes of the humidity level from 30 to 80 %, and then investigating energy harvesting performance in the optimal measurement condition as seen below. We found that the level of humidity did not influence the harvesting outcome.

Supplementary Fig. 10. Output voltage generated by an optimal CCTO harvester performance with the changes in humidity level as measured at bending strain of 0.77% and bending frequency of 2.90 Hz.

(4) Reliability of the harvesting performance after poling

We also ensured the reliable harvesting performance for the optimized sample after poling with the extended bending operations up to 11,000 cycles as seen below.

Supplementary Fig. 14 Stability evaluation of the output voltage over 11,000 cycles for the 4.3 mTorr sample poled at 120 kV cm^{-1} , measured at the bending strain of 0.77% and the bending frequency of 3.10 Hz.

The following descriptions correspond to the reliability evidence in the text:

“We also confirmed the stability of the harvesting performance by continuing bending up to 11,000 cycles, as shown in Fig. 2f where consistently stable output values were attained over the extended cycles at 2.73 Hz for the ~ 497 -nm-thick film. Supplementary Fig. 7 shows another case of the harvesting stability at a higher bending frequency of 2.90 Hz for the identical sample, confirming that the physical integrity is well preserved during repetitive bending operations. Chemical stability of the amorphous CCTO films was evaluated in terms of changes in chemical states of ions after exposing the films in ambient atmosphere for 15 days as compared in the XPS spectra of Supplementary Fig. 8 before and after the exposure. As expected, no noticeable chemical changes were observed with no degradation in the harvesting performance after the exposure period as presented in Supplementary Fig. 9. As another effort, the harvesting performance was monitored with the changes in humidity level from 30 to 80 % as seen in Supplementary Fig. 10, suggesting that the optimal harvester device is quite stable at the humid atmosphere.”

(in page 9, line 6)

“The reliability of the optimal harvester is confirmed by ensuring the consistent harvesting performance up to 11,000 cycles as seen in Supplementary Fig. 14.”
(in page 16, line 16)

3. The author has provided the PENG device output with and without the PDMS layer. In the case of PDMS, the output peaks are more stable. The author needs to clarify what encapsulation was used for the device without a PDMS layer, as without any antistatic protection, the external impact layer influences the output.

REPLY) For the experiment of the harvesting devices with and without the PDMS layer, we used an identical polyimide(PI) tape for the final encapsulation with external Cu wires for both the cases, meaning that the identical device structure except the usage of the PDMS layer (or not) was used for the evaluation.

To confirm no effect of the PI encapsulation, we made extra harvester device using an Ag-based conductive epoxy (8331S, MG Chemicals) for the attachment of Cu wires (via bonding) without using the PI. The testing results were basically identical between the usages of the PI tape and conductive epoxy as seen below, meaning that the PI encapsulation might not contribute to the harvesting outcomes.

Comparison in energy harvesting performance between the usages of PI tape and conductive epoxy for the attachment of Cu wires: (a) output voltage and (b) output current

4. The author needs to justify the drastic increment of the piezoelectric, whether the randomly oriented dipoles are aligned and contributing to enhancement or decrement of the piezoelectric performance.

REPLY) The drastic increases in harvesting performance were observed with increasing oxygen partial pressure, which induced nearly stoichiometry compositions with the highest oxygen pressure. As seen below and mentioned in the 1st revision, applying poling field did not create dramatic increases in harvesting outcomes. The output voltage and current values were raised to 38.7 V and 1,238 nA, respectively, at the maximum field of 120 kV cm⁻¹, which correspond to the increments by ~76% and ~37% compared to the values (22.0 V and 906 nA) attained without poling. Unlikely to the crystalline CCTO, all the TiO₆ octahedra may not only be connected via apex-to-apex,

but also via edges and faces in the amorphous state. Accordingly, the alignment of dipoles along the poling direction may not be great as much as observed in the crystalline perovskite.

Fig. 4 a,b Output voltage and current obtained with the ~497-nm-thick CCTO harvester after applying the electric fields, which were measured under the optimal conditions of 0.77% and 3.10 Hz. **c** Plots of peak voltage and current with increasing poling field.

To visualize the dipole alignments with poling field, we provided the PFM phase image obtained by applying +10 and -10 V consecutively in the designated boxes as shown below (as Fig. 3e). The clear changes in contrast suggest that the domain alignments (or reversal) happened with the poling field in the amorphous film.

Fig. 3 e PFM phase image characterized by applying consecutive biases of +10 and -10 V in the designated boxes.

As another way of visualization, the following plots of the changes in distribution of phase angles with positive and negative poling demonstrate clearly the domain reversals in the amorphous films (which was extracted from the image of Fig. 3e)

Distribution of PR phase angles for (a) unpoled, (b) positively poled, and (c) negatively poled cases, which were extracted from the PFM phase image of Fig. 3e.

We designed Fig. 3 to demonstrate the origin of the enhanced piezoelectricity with the higher oxygen pressure by chasing the experimental evidences using PFM, laser interferometry, four-point bending unit (for $-e_{31,eff}$), ϵ_r dispersion with frequency, and impedance plots. In fact, we believe that the robust enhancements in piezoelectric harvesting performance themselves are directly related to improved piezoelectricity of the amorphous films.

5. The author has provided the influence of the oxygen pressure, thickness, strain, bending strain, and bending frequency. The linear increment of output is observed in all the parameters. However, is there any standard common correlation between these factors contributing to performance enhancements?

REPLY) Unfortunately, as we believe, there is no standard correlation between these factors because the factors affect different aspects in generating harvesting performance. For example, oxygen partial pressure concerns the film's chemical compositions (towards the stoichiometry with the higher oxygen partial pressure), while bending strain provides different levels of input mechanical source (a higher bending strain induces better harvesting outcomes). We double-checked the literature about the standard common correlations between these factors in any harvesting system, but we could not find any clue.

6. The material and concept of piezoelectricity are described with various experimental tools. The author must provide insight into the fabrication of high-performance PENG and some methods to reduce the higher poling voltage for a sustainable and reliable energy harvesting technique.

REPLY) We agree that reducing the poling field is meaningful with reliable harvesting performance. Our harvesting performance is still very good even without the additional poling process: 22.0 V and 906 nA, compared to the values of 38.7 V and 1,238 nA after poling at of 120 kV cm^{-1} . In most of the reported piezoelectric (crystalline) thin films, the poling process has been inevitably applied for the piezoelectric enhancement via domain orientation, mostly in the film systems on Si substrates. Note that Supplementary Table 3 demonstrated the harvesting results for other piezoelectric thin films poled at certain fields, which are far less than our unpoled values.

In this regard, we added a short description as following:

“It is worth mentioning that our harvesting performance of 22.0 V and 906 nA is still viable even without the poling process. There is no such outstanding performance for piezoelectric thin films particularly without applying the poling field.”

(in page 16, line 17)

[**NOTE:** We were initially surprised with our promising harvesting performance of the amorphous CCTO films because there was no such case. We repeatedly performed the harvesting experiments whether this performance is true or not while chasing the potential origins of the enhancements. We believe that we used all available analytical tools to support our findings even if it may not be fully competent. Nonetheless, the excellent harvesting performance is true, as we strongly believe. Probably, some following studies on different amorphous perovskites may need for further verification of the harvesting potentials in amorphous states]

Reviewer #3 (Remarks to the Author):

The authors have addressed most of the concerns, and the quality of the manuscript has been improved with the additional experiments and analyses.

REPLY) We appreciate your positive comment.

REVIEWERS' COMMENTS

Reviewer #2 (Remarks to the Author):

The author has addressed the comments satisfactorily, so the overall quality of the manuscript is enhanced and can be accepted in its current state.